



**1** **Real-world observations of ultrafine particles and reduced nitrogen in**

**2** **commercial cooking organic aerosol emissions**

**3** Sunhye Kim[1], Jo Machesky[2], Drew R. Gentner[2], Albert A. Presto[1]

**4** [1.] Department of Mechanical Engineering and Center for Atmospheric Particle Studies, Carnegie
**5** Mellon University, Pittsburgh, Pennsylvania, United States
**6**
**7** [2.] Department of Chemical & Environmental Engineering, Yale University, New Haven,
**8** Connecticut 06511, United States
**9**
**10** Correspondence: Albert A. Presto (apresto@andrew.cmu.edu)

**12** **Abstract**

**13** Cooking is an important but understudied source of urban anthropogenic fine particulate matter

**14** ($PM_{2.5}$). Using a mobile laboratory, we measured PM size and composition in urban restaurant

**15** plumes. Size distribution measurements indicate that restaurants are a source of urban ultrafine

**16** particles (UFPs, particles <100 nm diameter), with a mode diameter <50 nm across sampled

**17** restaurants and particle number concentrations (PNC, a proxy for UFPs) that were substantially

**18** elevated relative to the urban background. The majority of observed PM was organic aerosol

**19** (OA) by mass. Aerosol mass spectra show that while emissions from most restaurants were

**20** similar, there were key mass spectral differences. All restaurants emit OA at *m/z* 41, 43, and 55,

**21** though the composition (e.g., the ratio of oxygenated to reduced ions at specific *m/z*) varied

**22** across locations. All restaurant emissions included reduced nitrogen species detected as $C_xH_yN^+$

**23** fragments, making up ~15% of OA mass measured in plumes, with reduced molecular

**24** functionalities (e.g., amines, imides) that were often accompanied by oxygen-containing

**25** functional groups. The largest reduced nitrogen emissions were observed from a commercial



bread bakery (i.e., 30-50% of OA mass), highlighting the marked differences between restaurants
and their importance for emissions of both urban UFPs and reduced nitrogen.
**Introduction**

Concentrations of most air pollutants, including fine particulate matter ($PM_{2.5}$) and

ultrafine particles (UFPs; particles with diameter <100 nm), are typically higher in urban areas
compared to rural or suburban areas (Cheng et al., 2019; Chow et al., 2006; Lenschow et al.,
2001; Louie et al., 2005; Renzi et al., 2021; Wang et al., 2020). Elevated urban concentrations
lead to higher human exposure, and in turn, contribute to the health impacts of air pollution.
$PM_{2.5}$ exposures are associated with cardiovascular disease, lung cancer, and asthma and
contribute to up to 100,000 deaths annually in the US (Castillo et al., 2021). Although health
effects of UFP exposure are less extensively studied compared to $PM_{2.5}$ (Schraufnagel, 2020) and
are an area of ongoing research, there is growing evidence that UFPs can enhance acute health
effects because of their small size and high surface area (Ali et al., 2022; Ibald-Mulli et al., 2002;
Kwon et al., 2020).

The $PM_{2.5}$ and UFP concentration enhancements in many urban areas are strongly

influenced by anthropogenic emissions (Apte et al., 2017; Li et al., 2018; Mohr et al., 2011; Saha
et al., 2019). Among a wide variety of contributing sources, two notable urban sources are
mobile sources (e.g., motor vehicles) and cooking. These two sources contribute to urban
enhancements relative to the non-urban areas and to intra-urban spatial variations in $PM_{2.5}$ and
UFP concentrations (Klompmaker et al., 2015). In prior work, mobile sources and cooking
emissions have led to neighborhood-scale enhancements of ~0.5-1 µg m$^{-3}$ of $PM_{2.5}$ in North





American cities and a factor of two enhancement in UFPs (Rose Eilenberg et al., 2020; Song et
al., 2021b).
Motor vehicle emissions are well studied and have seen dramatic reductions as a result of
effective regulations on PM emissions across Europe and the US (Font et al., 2019; Keuken et
al., 2012). In contrast, there has been less attention to cooking sources as contributors of PM and
UFP emissions. As such, there have been fewer direct measurements and regulations dedicated
to cooking-related emissions, including everyday sources such as restaurants and home kitchens.
For comparison, two studies conducted in Pasadena, California revealed that organic $PM_{2.5}$
attributed to cooking decreased from approximately 2.4 µg/m$^3$ to 1.2 µg/m$^3$ between 1982 and
2010, while the contribution from traffic sources dropped from about 6.8 µg/m$^3$ to 0.82 µg/m$^3$
(Hayes et al., 2013; Schauer et al., 1996). This means that while total $PM_{2.5}$ and vehicular-related
primary $PM_{2.5}$ have decreased, the fraction of urban $PM_{2.5}$ attributed to cooking has increased.
Aerosol mass spectrometry (AMS) measurements worldwide further indicate the
importance of cooking PM. Factor analysis of AMS using PMF (Positive Matrix Factorization)
data routinely identifies a Cooking Organic Aerosol (COA) factor that accounts for 6 - 25% of
the total organic aerosol (OA) in urban environments. Specifically, a measurement study in
Athens and Patras, Greece, showed that the COA contribution increased to 75% of organic $PM_1$
during mealtime in Patras (Florou et al., 2017). While the COA factor is routinely identified,
there can be significant variation in its composition from city to city (Bozzetti et al., 2017;
Crippa, El Haddad, et al., 2013; R. Hu et al., 2021; X.-F. Huang et al., 2010; Lee et al., 2015;
N. Pandis et al., 2016; Rogge et al., 1991a; Sun et al., 2012).
Many potential factors could produce variability in the composition and size distribution
of cooking PM.  While the UFPs from cooking can contribute to ~ 80% of the total particle



number concentrations indoors (Wan et al., 2011), there are a lot of factors—such as indoor-
outdoor air exchange rates (Wallace et al., 2004) and types of cooking oils used (Torkmahalleh
et al., 2012)—that can determine the size distribution of particles as well as the $PM_{2.5}$
concentrations from cooking activities. There is some evidence that the chemical composition of
cooking emissions may vary with the cooking style and the food cooked (Omelekhina et al.,
2020; Reyes-Villegas et al., 2018a; Takhar et al., 2019). For example, the cooking temperature,
ingredients, and methods used can alter chemical pathways that lead to the generation of
nitrogen-containing functional groups, including amides, within COA (Ditto et al., 2022).
Multiple studies found that nitrogen composition has been observed while charbroiling (Rogge et
al., 1991a) or deep-frying hamburgers (Reyes-Villegas et al., 2018b; Rogge et al., 1991a).
Masoud et al., (2022) found that nitrogen-containing compounds contributed 12-19% of the
signal measured by a chemical ionization mass spectrometer for emissions from typical in-home
cooking. Overall, this variability across diverse cooking styles and conditions is relevant but
poorly understood. This implies a significant need for real-world measurements to characterize
and understand particle size and composition of cooking emissions in urban environments.

This study aimed to characterize cooking emissions from real-world restaurant sources in

the US. We used a mobile laboratory to measure cooking emissions from nine restaurants in
Pittsburgh, PA and Baltimore, MD. Four of these restaurants were visited twice, making for a
total of thirteen cooking sites. Several analytical instruments, including an AMS and FMPS (Fast
Mobility Particle Sizer), were used at each site for online measurements, with supplemental PM
collection on Teflon filters for offline analysis. The measurements are used to examine variations
in UFP concentrations and cooking OA composition measured outside of restaurants with a



focus on contributions from reduced nitrogen components across restaurant sites visited during
the field campaign.

**2. Methods**
*2.1 Measurement locations*
**Table 1.** Summary of restaurant locations and concentration enhancements measured in the
cooking emission plumes. Several restaurants were sampled on two separate days, as indicated
by the number following the restaurant identifier. AMS high-resolution analysis of mean OA
enhancement (CE=1), mean BC enhancement from aethalometer, Mode $D_p$ (nm), mean $f_{41}$ (the
fraction of mass-to-charge ratio at 41 to the total organic mass signal), $f_{43}$, and $f_{55}$.

| | City | Mean $\Delta$ OA ($\mu g/m^3$) | Mean $\Delta$ BC ($\mu g/m^3$) | Mode $D_p$ (nm) | $f_{41}$ | $f_{43}$ | $f_{55}$ |
|---|---|---|---|---|---|---|---|
| Island Cuisine | Pittsburgh | 65 | 0.83 | 17 | 0.068 | 0.054 | 0.094 |
| Pizza | Pittsburgh | 100 | 3.2 | 29 | 0.070 | 0.058 | 0.096 |
| BBQ | Baltimore | 1.2 | 0.38 | 11 | 0.061 | 0.058 | 0.070 |
| Café | Baltimore | 2.3 | 0.35 | 8.1 | 0.044 | 0.082 | 0.043 |
| Beef | Baltimore | 15 | 4.2 | 11 | 0.082 | 0.074 | 0.10 |
| Diner 1 | Pittsburgh | 77 | 1.4 | 11 | 0.065 | 0.044 | 0.078 |
| Diner 2 | Pittsburgh | 84 | 2.0 | 11 | 0.078 | 0.054 | 0.092 |
| Bakery 1 | Baltimore | 12 | 0.091 | 8.1 | 0.011 | 0.023 | 0.003 |
| Bakery 2 | Baltimore | 4.6 | 0.41 | 8.1 | 0.053 | 0.048 | 0.003 |
| Fast Food 1 | Baltimore | 1.7 | 1.4 | 29 | 0.030 | 0.064 | 0.024 |
| Fast Food 2 | Baltimore | 3.8 | 0.36 | 11 | 0.053 | 0.048 | 0.013 |
| Bar/Restaurant 1 | Baltimore | 69 | 2.4 | 11 | 0.086 | 0.066 | 0.10 |
| Bar/Restaurant 2 | Baltimore | 140 | 5.0 | 26 | 0.076 | 0.076 | 0.12 |


Field samples were collected from 13 visits to 9 urban cooking sites in Pittsburgh and

Baltimore during July and August 2019 (Table 1). At each location, we parked a mobile
laboratory near the restaurant's exhaust plume (SI Fig. 1). The selected restaurants represent a
mix of accessible locations with visible emission plumes or exhaust vents. The sampling inlet on
the mobile laboratory was typically within a few meters of the exhaust vent. However, this
distance varied due to several uncontrollable external factors, such as the availability of parking



and the height of the restaurants' exhaust vents. As a result, the measured emission plumes went
through varying degrees of dilution before reaching our sampling inlet.

Several of the restaurants were sampled on multiple visits to examine day-to-day

variations in emissions. These variations could be due to differences in activity (e.g., how many
customers ordered food), the type of food ordered, and differences in dilution conditions. Each
visit to a restaurant site lasted approximately 30-60 minutes. The sampling periods targeted
expected times for lunch (~11 am – 1 pm) and dinner (~6 – 8 pm).

*2.2 Mobile laboratory and measurements*

Instruments were loaded into a gasoline-powered mobile laboratory. At each location, we

oriented the mobile laboratory so that the vehicle exhaust was located downwind of the sample
inlet to minimize self-contamination from the vehicle exhaust.

We use total particle number concentration (PNC) as our proxy for UFPs. Particle

number counts were measured by a MAGIC$^{TM}$ water CPC (Moderated Aerosol Growth with
Internal water Cycling Condensation Particle Counter, Aerosol Devices Inc, Model
MAGIC200P). MAGIC CPC uses water condensation to enlarge particles through a 3-
temperature stage growth tube. The enlarged particles are counted with a laser sensor up to
400,000 particles cm$^{-3}$ with a particle size range between 5 nm and 2.5 µm in diameter (Hering et
al., 2019). Saha et al., (2019) previously observed that the MAGIC CPC undercounts relative to a
butanol CPC. Thus, the raw CPC output was adjusted using a correction factor determined from
the co-location of the MAGIC CPC with a TSI 3772 butanol CPC.

Particle size distributions and total number concentrations were measured with FMPS

(Fast Mobility Particle Sizer, TSI Inc, Model 3091) for particles with diameters from 6.04 nm to



523.3 nm. The FMPS reported systematically lower particle counts than the MAGIC CPC (factor
of 3.5, SI Section 2 and Fig. S2). FMPS data were utilized in lieu of the CPC data due to high
particle number concentrations in restaurant plumes that exceeded the upper counting limit of the
CPC (400,000 particles $cm^{-3}$), resulting in error flags. To ensure consistency with the MAGIC
CPC, all FMPS data were corrected by integrating the FMPS size distribution, which was scaled
by the FMPS:CPC ratio.

A High-Resolution AMS (HR-AMS, Aerodyne), which measures non-refractory particles

with a diameter less than 1 µm ($NR-PM_1$), was used to identify mass spectra of PM components
(Organics, $NH_4^+$, $NO_3^-$, $SO_4^{2-}$, and $Cl^-$) in real-time. Squirrel (SeQUential Igor data RetTriEvaL)
toolkit 1.62G and Pika (Peak Integration by Key Analysis) toolkit 1.22G in Igor Pro
(Wavemetrics, Lake Oswego) were used for the HR-AMS data analysis. For the baseline and
peak fitting correction procedures of the HR-AMS data, the high-resolution range of *m/z* (mass-
to-charge ratios) 12 to 140 was selected. All AMS analysis presented here assumes a collection
efficiency (CE) of one.

An aethalometer (Magee Scientific, Model AE33), CO analyzer (Teledyne API T300),

and $CO_2$ analyzer (LiCor LI-820, Biosciences) measured black carbon (BC), CO, and $CO_2$
concentration, respectively.

$PM_{2.5}$ samples were collected at ~70 L/min on 47 mm PTFE membrane filters (47 mm,

2.0 µm pores, Tisch Scientific) through a separate inlet mounted close to the online
instrumentation inlet outfitted with a cyclone (2.5 µm cut point with a flow rate of 92 LPM,
URG-2000-30EH, URG cyclone). At each restaurant site where plumes were observed via AMS,
a filter sample was collected for at least 30 minutes and Table S3 shows details for each filter
sample. Filter samples were transported on ice packs from the mobile lab and kept in sample



storage freezers. Additional filter collection details can be found in the Supporting Information.
All samples were analyzed via liquid chromatography (LC) using an Agilent Infinity LC and an
Agilent Poroshell 120 SB-Aq reverse-phase column (2.1×50 mm, 2.7 µm particle size). The LC
was coupled to an electrospray ionization (ESI) source, operated in positive and negative modes
for each sample, and connected to a high-resolution mass spectrometer (Agilent 6550 Q-TOF).
These instruments were operated following previously described methods (Ditto et al., 2018,

2020).

Selected samples showing unique AMS spectra with nitrogen-containing compounds

underwent further analysis via MS/MS (tandem mass spectrometry) with the objective of
identifying the distribution of functional groups within the reduced nitrogen species that were
observed via LC-TOF, similar to prior work (Ditto et al., 2020, 2022). LC-TOF mode data
processing and QC/QA have previously been described (Ditto et al., 2018), and details of
compound selection for MS/MS analysis in this study can be found in the Supporting
Information (Section S3). MS/MS spectra analysis used SIRIUS with CSI:FingerID for
molecular structure prediction (Dührkop et al., 2015, 2019), and the APRL Substructure Search
Program was used for functional group identification from the predicted SMILES formula for
atmospherically-relevant groups (Ruggeri & Takahama, 2016). Further details on LC-MS/MS
analysis, processing, and associated limitations of ESI and MS/MS spectra analysis can be found
in Ditto et al., (2020), with brief comments on relevant SIRIUS updates in the Supporting
Information (Section S3).







## 3. Results and Discussion

### 3.1 Typical measurements of restaurant emission

Figure 1 demonstrates observations collected during a typical sampling day via the mobile lab in Baltimore. On this day, the mobile laboratory was initially (~15:36 – 16:49 EDT) parked in an urban park, here noted as background. Sampling was then conducted on-road, driving on various streets in urban Baltimore, from 16:49 to 18:20. At 18:20; the mobile laboratory was parked outside a restaurant (Bar/Restaurant 2).

The data in Figure 1 exemplify clear variations in pollutant concentrations between the background, on-road, and restaurant portions of sampling. In general, concentrations were the lowest and least variable in urban background locations and the highest and most variable for the restaurant sampling periods.

Nearby vehicles likely impacted the measured concentrations during the on-road sampling period, thus differentiating it from the background period, where direct observations of on-road emissions were minimal. Concentrations of CO, $CO_2$ (Fig. 1a), organic aerosol (OA), black carbon (BC, Fig. 1b), and particle number (Fig. 1d) were all elevated in the on-road samples compared to the urban background. ΔOA and ΔBC were calculated by subtracting the background concentration from the measured OA or BC mass concentration. The background concentration is defined as the 5[th] percentile of data collected on each sampling day (listed in Table S1).

The mean organic aerosol concentrations are 5.8 $\mu g/m^3$ (ΔOA: 2.46 $\mu g/m^3$) during on-road sampling versus 4.2 $\mu g/m^3$ (ΔOA: 0.85 $\mu g/m^3$) in the urban background (Fig. 1b). Similarly, the BC concentration was 0.5 $\mu g/m^3$ higher on-road than in the urban background, and PNC was approximately a factor of three higher on-road than in the urban background. These





enhancements in organic aerosol, black carbon, and PNC are broadly consistent with
enhancements with those seen in high-traffic areas by our previous sampling in Pittsburgh and
Oakland (Saha et al., 2020; Shah et al., 2018).

In addition to the overall increase in pollutant concentrations on-road, there are

occasional, coincident spikes in CO, BC, OA, and PNC during the on-road sampling. The
particle size distribution also changes during these spikes (Fig. 1c), with higher concentrations of
particles in the 20-100 nm size range. These are likely plumes from high-emitting vehicles,
potentially diesel trucks and buses (Dallmann et al., 2013; Tan et al., 2016).

The highest and most variable concentrations are observed in the restaurant plume. In this

near-source environment, organic aerosol concentrations averaged 146 µg/m$^3$. This is 35 times
higher than the urban OA background. Particle number counts were also 35 times higher in
concentration than background levels. CO, CO2, and BC enhancements were also observed
when the mobile lab was parked near the restaurant. The enhancement of CO was 5.9 times the
background, $CO_2$ and BC were 1.15 and 5.42 times higher, respectively.

During the restaurant sampling period, there are several clear and concurrent spikes in

OA (Fig. 1b) and particle number count (Fig. 1d). These seem to be associated with specific
events, such as preparing a customer's new order (restaurant kitchens had varying activity levels
during the sampling periods). The size distributions in Figure 1c show that these emissions span
a wide range in particle size, from <10 nm up to a few hundred nm, demonstrating that
restaurants may be a source of urban ultrafine particles.



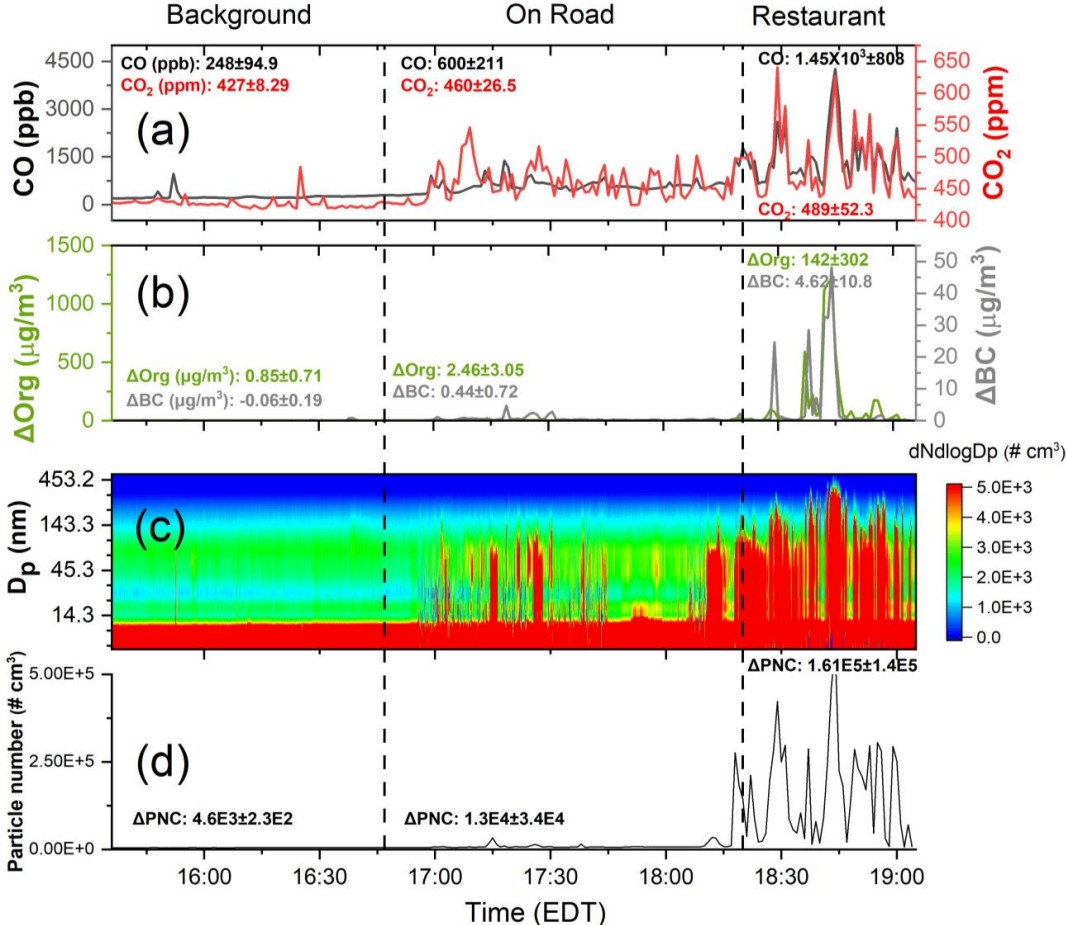

**Figure 1.** Urban background, on-road, and restaurant plumes observed during a typical sampling
day (Bar/Restaurant 2) in Baltimore, showing: (a) CO and $CO_2$, (b) background corrected
organic aerosol (OA) and black carbon (BC) concentrations, (c) particle size distribution from
FMPS, and (d) background-corrected total particle number concentrations. All concentrations
were significantly higher and more variable in restaurant emissions plume than in the urban
background or on-road period. Numbers in (a), (b), and (d) indicate the mean ± standard
deviation for each sampling period.

While average BC concentrations were about a factor of five higher than background

during the restaurant sampling period, BC seems to be a relatively smaller component of PM
emissions from the restaurant. The OA/BC ratio in the urban background and on-road sampling




periods was ~4. In the restaurant plume, the mean OA/BC ratio was 28. Despite occasional
periods of very high BC concentrations reaching up to 58 µg/m$^3$, the OA/BC ratio during the
spike was 230 (Fig. S3). Other PM components (e.g., sulfate and nitrate) show no discernable
enhancement during the restaurant sampling period (Fig. S4). This indicates that the PM
emissions from the restaurant were dominated by organic aerosol.

We also observed elevated concentrations of CO and $CO_2$ in the restaurant exhaust. We

do not have information about each restaurant's cooking practices or fuels (i.e., whether the
restaurants used natural gas or electricity). Jung & Su (2020) showed that food cooking emits
CO, so the CO spikes observed here may also be from the food rather than fuel combustion.
Other recent measurements in Pittsburgh by (Song et al., 2021a) also showed enhancements in
CO during mealtimes in a restaurant-rich area.

*3.2 Summary of organic aerosol enhancements at restaurant sites*

Enhancements in OA as a result of emissions from restaurants were similarly observed

across all other sampling sites that we visited. Figure 2 is a box-plot visualization of the OA
enhancement (ΔOA) for each restaurant visit. The data are split into two main groups for visual
clarity: high concentration (mean ΔOA > 50 µg m$^{-3}$, Fig. 2a) and low concentration (mean ΔOA
< 30 µg m$^{-3}$, Fig. 2b).



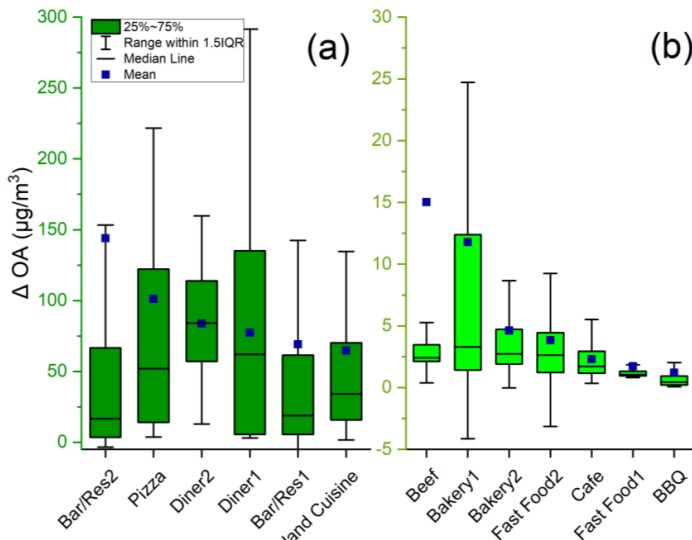

**Figure 2.** Organic aerosol enhancement (ΔOA) at each restaurant site with (a) high (mean ΔOA > 50 μg/m$^3$) and (b) low (mean ΔOA < 30 μg/m$^3$) enhancements grouped in each for comparison. The sample names in (a) and (b) are ordered by decreasing mean concentration.

There is significant variability in measured ΔOA between and within each restaurant (Fig. 2 and Fig. S4). For nearly every location sampled, the emissions varied over time, as shown in Figure 1, and this contributes to wide interquartile ranges (IQRs) in Figure 2. It also means that at nearly every restaurant, there were periods when the concentration was near the urban background level, as indicated by the whiskers reaching (or even going slightly below) zero.

The temporal variability of the concentrations measured at each restaurant contributed to an upward skew in ΔOA, with a mean concentration greater than the 75th percentile at many locations. This suggests that the measurements were dominated by short, intense bursts of emissions rather than sustained high concentrations. Visualizations of this trend are noticeable in Figure 1b, where there is a large spike in emissions so that OA goes above 1000 μg/m$^3$ for several minutes. The temporal variability seems to be associated with the quantity of cooking that spikes amid busy mealtimes.




Four restaurants were sampled on multiple days (Bar/Restaurant, Fast Food, Bakery, and
Diner). While there were day-to-day differences in the mean $\Delta$OA at each location, each of these
locations fell into the same group (i.e., $\Delta$OA < 30 $\mu$g m$^{-3}$ or $\Delta$OA > 50 $\mu$g m$^{-3}$) on both sampling
days. This suggests that the day-to-day variations in emissions are smaller than within-day
emissions for each location and that high-emitting restaurants are consistently high emitters.
However, due to the limitation of a single visit to each sampling location during the campaign, it
may be challenging to conclusively ascertain that the classification assigned to the sampled
restaurants is not indicative of all similar cooking operations.

*3.3 OA composition across restaurants*

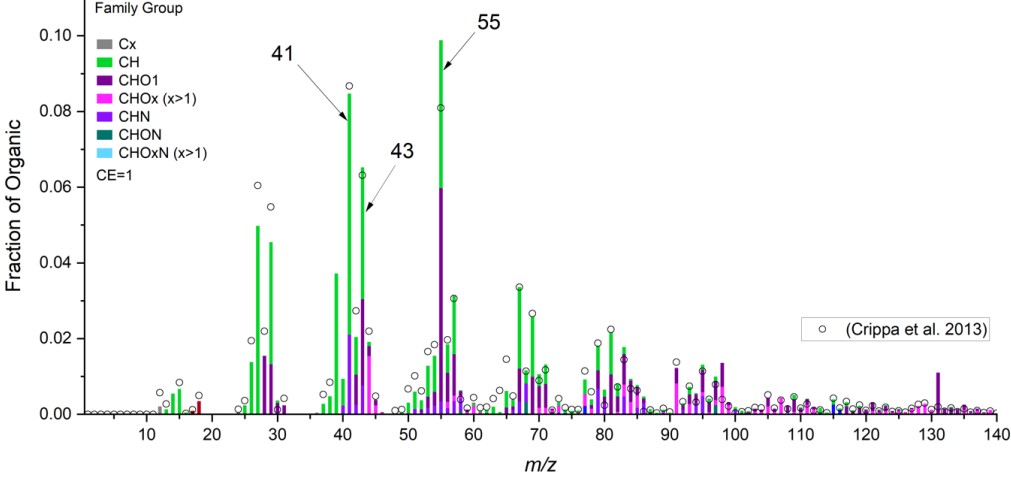


**Figure 3.** Example mass spectrum from Bar/Restaurant 1 in this study and comparison with the
COA mass spectrum from prior PMF work. High-resolution mass spectra are grouped into sticks
of the unit mass resolution, and the coloring of each stick represents the mass fraction belonging
to different chemical families.


In this section, we compare the composition of cooking OA across the restaurants and to
previous laboratory measurements and ambient factor analysis. Figure 3 shows the mean mass



spectrum of OA measured at Bar/Restaurant 1 in Baltimore; mass spectra from three additional
restaurants are shown in Figure S5. The mass spectrum contains a mixture of hydrocarbon
($C_xH_y$) and oxygenated ($C_xH_yO$) ions. This is consistent with the composition of cooking OA,
which is often dominated by long-chain fatty acids from heated cooking oils and from meat
cooking (Crippa, DeCarlo, et al., 2013; D. D. Huang et al., 2021a; Liu et al., 2017; Mohr et al.,
2009; Takhar et al., 2019; Z. Zhang et al., 2021). Several lab experiments from seed oil cooking
detected fatty acids or degradation fragments such as *n*-alkanoic acid, *n*-alkenoic acid, oleic acid,
and carbonyls (Allan et al., 2010; Liu et al., 2018; Schauer et al., 2002). Unlike oils, which are
entirely comprised of fats, meats contain proteins and fats, although the composition can vary
depending on the type of meat. Cooking meat generally emits cholesterol and fatty acids like
palmitic acid, stearic acid, and oleic acid (Rogge et al., 1991a; Schauer et al., 1996), which have
all been used as chemical markers of meat cooking emissions. This mixture of hydrocarbon and
oxygenated ions is also identified in PMF factor analysis of ambient datasets, as indicated by the
mass spectrum from Crippa, DeCarlo, et al., (2013) shown in Figure 3.

The most abundant peaks in the mass spectrum were at *m/z* 41 (mostly $C_3H_5^+$), 43

($C_2H_3O^+$ and $C_3H_7^+$), and 55 ($C_3H_3O^+$ and $C_4H_7^+$). These peaks have been used as COA markers
for tracing cooking sources in previous studies (Allan et al., 2010; Dall'Osto et al., 2015;
Kaltsonoudis et al., 2017; Mohr et al., 2009). Table 1 summarizes the mean contribution ($f_{41}$, $f_{43}$,
and $f_{55}$) at these *m/z* to each restaurant's overall OA mass spectrum.

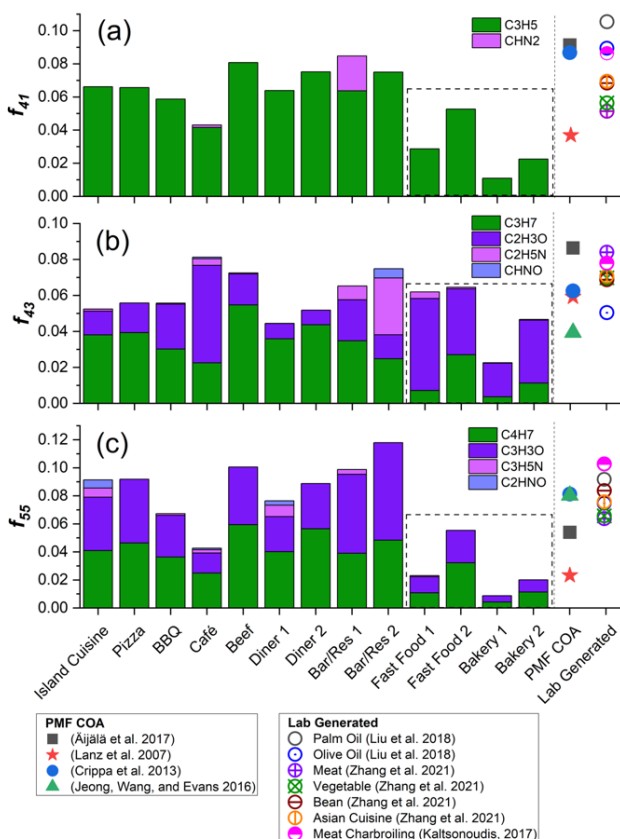


**Figure 4.** Fraction of (a) m/z 41, (b) 43, and (c) 55 to the total organic aerosol concentrations and comparison to COA mass spectra from prior PMF studies (Äijälä et al., 2017; Crippa, DeCarlo, et al., 2013; Jeong et al., 2016; Lanz et al., 2007) and laboratory-generated cooking emissions (Kaltsonoudis et al., 2017; Liu et al., 2018; Z. Zhang et al., 2021). Only $f_{43}$ and $f_{55}$ were shown in (Jeong et al., 2016) ($f_{41}$ was not provided in the paper). Fast Food and Bakery samples are grouped in a box as they showed lower abundances of these common cooking marker fractions ($f_{41}$, $f_{43}$, and $f_{55}$).

320    Figure 4 compares $f_{41}$ (OA mass fraction at *m/z* 41), $f_{43}$, and $f_{55}$ across the restaurants

321   sampled here to previously published COA mass spectra. We compared two types of previous

322   studies: COA mass spectra derived from factor analysis of ambient data using PMF and

323   laboratory measurements of cooking emissions. The laboratory measurements shown here





include a combination of heating palm and olive oils (Liu et al., 2018) and various cooking
experiments using meats (chicken and pork), vegetables, beans, and Asian cuisine (Kaltsonoudis
et al., 2017; Z. Zhang et al., 2021).

For $m/z$ 41, our data were dominated by the hydrocarbon ion ($C_3H_5^+$), which was

approximately 4-8% of OA mass for most of the restaurants. The exceptions were Fast Food 1
and the two samples collected at the Bakery location. These had lower $f_{41}$ (1-5%) and are shown
inside the dashed box. $f_{41}$ fractions from our study were generally lower than from the PMF
COA factors. Three of the four COA factors have $f_{41}$ of ~9% (Äijälä et al., 2017; Crippa,
DeCarlo, et al., 2013; Jeong et al., 2016). The COA factor from Lanz et al., 2007 is 4% and is
lower than most of the restaurants we sampled here. There is a wide range in $f_{41}$ from the
laboratory experiments. The two oil heating experiments (palm and olive oil, Liu et al., 2018)
generated higher $f_{41}$ than most of our measurements (8-10%). There was a wider range in $f_{41}$ for
food cooking experiments (5-8%), and there is a strong overlap with our measurements.

For $f_{43}$ and $f_{55}$, both oxidized (e.g., $C_2H_3O^+$ and $C_3H_3O^+$) and hydrocarbon (e.g., $C_3H_7^+$

and $C_4H_7^+$) ion fragments showed significant contributions across the urban cooking sites. There
were also minor contributions from nitrogen-containing ions (e.g., $C_2H_5N^+$ and $C_2HNO^+$). Except
for Bakery 1, $f_{43}$ was ~5-8% in our measurements. However, there was variation in the relative
abundance of the hydrocarbon and oxygenated ions. For most sites, the contribution of the
hydrocarbon ($C_3H_7^+$) was larger than the oxygenated ion ($C_2H_3O^+$). However, the sites with low
$f_{41}$, Bakery and Fast Food 1, $m/z$ 43 fragments were mostly oxygenated (mean = 3.5%).

The mean $f_{43}$ in the PMF profiles was 6.3% with a range of 4-8.7%, which is similar to

the mean and range observed in our dataset. Similarly, the laboratory emissions data cluster




around $f_{43}$ of 8%, with slightly lower $f_{43}$ in the heated oil experiments. This is slightly higher
than the $f_{43}$ measured in the restaurant emissions.
The pattern in $f_{55}$ is similar to $f_{43}$; contributions are dominated by the hydrocarbon and
oxygenated ion, with minor contributions from N-containing ions. For most sites, including the
Bakery and Fast Food sites, the contributions of hydrocarbon and oxygenated ions at *m/z* 55 are
similar. The largest difference is that the Bakery and Fast Food sites have significantly lower $f_{55}$
(1-6%) than the other sites (4-12%). Additionally, for many of the sites, $f_{55}$ is larger than the
PMF factors and the laboratory experiments.
The variations in $f_{41}$, $f_{43}$, and $f_{55}$, as well as variations in the ratios between these *m/zs*,
may indicate the food cooked at the different restaurants. For example, $f_{41}$ seems to be larger
than $f_{43}$ for cooking emissions dominated by oil; this is the case in the oil heating experiments
from Liu et al. 2018 as well as from laboratory oil cooking emissions measured by Allan et al.
(Allan et al., 2010). Meat cooking emissions seem to have the opposite relationship, with $f_{43} >$
$f_{41}$. Both oil cooking and meat cooking have high $f_{55}$, and meat cooking may have $f_{55} > f_{43}$ (Mohr
et al., 2009).
For most restaurants sampled here (except Bakery and Fast Food), *m/z* 55 is the most
abundant signal in the aerosol mass spectrum. Additionally, $f_{41}$ is slightly higher than $f_{43}$ for
these sites. This suggests a mixture of meat and oil cooking at these locations. For Bakery and
Fast Food, $f_{43}$ is typically the most abundant ion, with $f_{41}$ exceeding $f_{55}$. This may suggest a
different mix of food being cooked, or a difference in the cooking style, however, there is
insufficient evidence in the mass spectra to conclusively explain the differences.



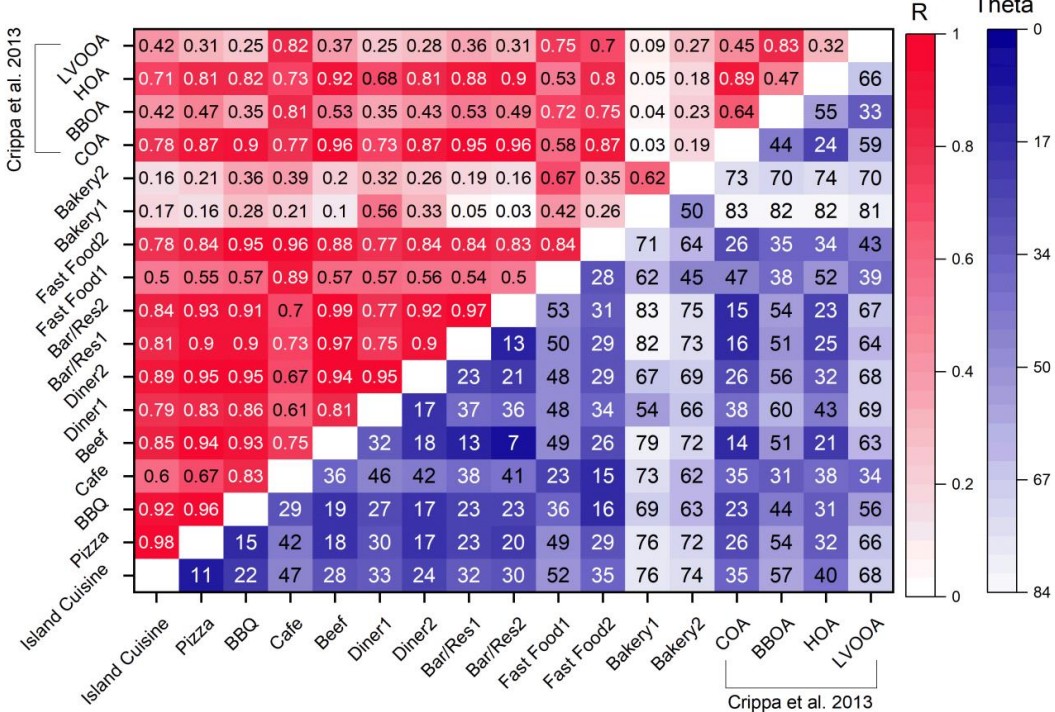


**Figure 5.** Comparison of the AMS UMR (unit mass resolution spectra) in two urban areas using correlation coefficients (R) and cosine similarity (θ, in degrees). R values close to 1 and θ values close to 0 mean strong correlations of mass spectra. Both R and θ values are presented such that darker colors correspond to higher similarity.


Figure 4 compares the cooking OA mass spectra for specific marker ions. Figure 5
compares the full cooking OA mass spectra. We use two metrics: the Pearson correlation (R) and
cosine similarity. The statistical approach, correlation coefficient R, has been widely used in
many studies, such as the analysis of air quality, to show an association between any two
variables (Devarakonda et al., 2013; Giorio et al., 2012; Kiendler-Scharr et al., 2009;
Raatikainen et al., 2010). Cosine similarity treats pairs of mass spectra as vectors and computes
the angle (θ) between them (Kaltsonoudis et al., 2017; Kostenidou et al., 2009). θ is a measure of
the similarities between two mass spectra, with a value of 0º, meaning that both spectra are



identical and θ>30° indicating considerable differences between the spectra. Cosine similarity is
more sensitive to smaller differences in mass spectra than R, as the correlation coefficient can be
dominated by ions with large abundance (Kaltsonoudis et al., 2017). Figure 5 also compares the
cooking emissions to PMF factors retrieved from Paris during winter (Crippa, DeCarlo, et al.,
2013) for biomass burning (BBOA), combustion emissions (HOA), and secondary OA
(LVOOA) obtained from the Jimenez Research Group website.
(http://cires1.colorado.edu/jimenez-group/AMSsd/).

Overall, the COA measured from most of the restaurants is similar. For most restaurants,

the R between mass spectra is larger than 0.8 and θ is less than 27°, suggesting that the mass
spectra are similar. Figures 3 and 4 show that the dominant ions in these mass spectra are at *m/z*
41, 43, and 55. The exceptions are the Bakery samples and, to a lesser extent, the Fast Food
samples. Bakery samples had R < 0.3 and θ > 50° when compared to most of the other
restaurants. There were day-to-day differences in the Fast Food mass spectrum, with one day
(Fast Food 1) being similar to other restaurants (R = 0.7-0.8, θ < 30°), and the other day (Fast
Food 2) having lower R and higher θ.  The following section discusses key mass spectral
differences in more detail.

There is also a high correlation of most restaurants with the COA PMF factor from

Crippa et al., (2013) (R > ~0.75, θ < ~30°). Correlations with BBOA and LVOOA are weaker as
these factors are characterized by dominant peaks at *m/z* 60 and 73 for BBOA and *m/z* 44 and 43
for LVOOA. There is a high R between our COA and the PMF HOA factor, which is
representative of primary combustion-related emissions. Even though *m/z* 41, 43, and 55 are
useful COA markers to resolve cooking-related factors, there are diverse sources of *m/z* 41, 43,
and 55. In general, there is a high correlation between HOA and COA because the major HOA



peaks like $m/z$ 55 and 57 are prominent in both factors (Milic et al., 2016; Sun et al., 2013; D.
Yao et al., 2021).

One key difference between HOA and COA is that the HOA mass spectrum is dominated

by hydrocarbon ($C_xH_y$), whereas the cooking OA has a mixture of hydrocarbon and oxygenated
ions, as shown in Figure 4. For example, $m/z$ 43 in HOA is almost entirely due to $C_3H_7^+$ (Ng et
al., 2010), whereas cooking OA contains both $C_3H_7^+$ and $C_2H_3O^+$ (Fig. 4). Similarly, for $m/z$ 55,
COA has contributions from both hydrocarbon ($C_4H_7^+$) and oxidized ($C_3H_3O^+$) fragments
(Canonaco et al., 2013; Lalchandani et al., 2021), whereas the reduced ion dominates HOA.
Lastly, while $m/z$ 55 and 57 are important signals for both COA and HOA, COA typically has $f_{55}$
$> f_{57}$, whereas HOA has the reverse (W. Hu et al., 2016; D. D. Huang et al., 2021a; Mohr et al.,
2009; Shah et al., 2018; Y. Zhang et al., 2015; Zhu et al., 2018).

*3.4 Cooking as a source of urban reduced nitrogen*

Cooking OA from all of the restaurant sites had a significant contribution from AMS ions

containing reduced nitrogen. The mean contribution of nitrogen-containing fragments to the total
cooking OA mass was 15.8% (median = 10.7%; Table S2). The bulk of these N-containing ions
(95% by mass) did not contain oxygen (Fig. S6), though oxygen could still be present on the
parent molecule prior to fragmentation. These $CxHyN^+$ fragments include $C_2H_5N^+$ ($m/z$ 43) and
$C_3H_5N^+$ ($m/z$ 55), shown in Figure 4. For example, the mass spectrum from Bar/Restaurant 1 in
Figure 2 has 9% CHN family peaks by mass, with significant contributions at $m/z$ 41 and 43. For
nearly all restaurants sampled here, the most abundant CHN group ion was $C_3H_8N^+$, with $f_{C3H8N}$
typically $> 1\%$.



Previous studies have reported the existence of nitrogen compounds or fragments from

cooking experiments. These nitrogen-containing compounds can originate from the food itself or
reactions with the types of gas used during cooking (Abdullahi et al., 2013). Reyes-Villegas et
al., 2018 measured gas- and particle-phase emissions and found 14 different nitrogen-containing
compounds using chemical ionization mass spectrometry. Rogge et al., 1991 measured amides in
cooking emissions, including palmitamide and steramide. Amides were also identified from both
Chinese cooking (Y. Zhao et al., 2007a) and Western-style cooking (Y. Zhao et al., 2007b) using
GC-MS. Ditto et al., (2022) recently demonstrated that amides can be formed from the reaction
of ammonia formed by amino acid thermal degradation with triglyceride ester linkages. In
contrast to the reduced nitrogen in our samples, these nitrogen-containing compounds, including
amides, have at least one oxygen in their formula.

The Bakery 1 and Bakery 2 samples had the largest contributions from reduced N. Figure

6 shows the aerosol mass spectrum from Bakery 1. The two most abundant ions in the mass
spectrum are $C_3H_8N^+$ (*m/z* 58) and $C_5H_{12}N^+$ (*m/z* 86); together these two ions make up ~48% of
the AMS-measured OA mass spectra. There is also a large contribution from $C_6H_{14}N^+$ at *m/z* 100.
The large abundance of these reduced N-containing peaks contributes to the low correlation
between the Bakery samples and other sites in Figure 5.



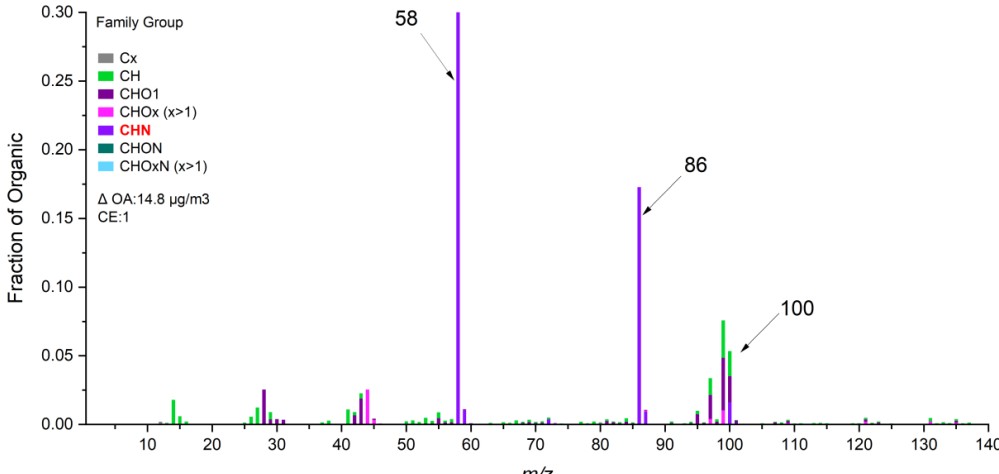

**Figure 6.** The aerosol mass spectrum from Barkery 1 with prominent peaks at m/z 58, 86, and 100 that are in the CHN family.

Though fast food sites have a lower correlation with other cooking sites in Figure 5, it is not primarily due to higher CHN levels like bakery samples. The most abundant signals of Fast Food 1 and Fast Food 2 were in the category of CHO and CH groups, where their sum accounts for 73.3 % and 82.0 % of the total mass, respectively. Two samples from Fast Food sites show moderate to slightly large proportions of CHN family peaks (14% and 7%) and $f_{C3H8N+}$ (2.15 and 2.33).

While the $C_3H_8N^+$ fragment has been observed in all of our cooking site data, there is almost no contribution of m/z 86 ($C_5H_{12}N^+$) and 100 ($C_6H_{14}N^+$) in our samples except for the two bakery visits (Table S2), which were collected adjacent to a large commercial bread bakery. It is thus possible that m/z 86 and 100 are more associated with commercial bakeries than restaurant cooking. The underlying source of the reduced nitrogen ions, especially m/z 86 and 100 observed at the bakery, is unknown. One potential source could be the use of azodicarbonamide ($C_2H_4N_4O_2$, ADA), which is used as an aging and bleaching ingredient in bread baking. To test





whether ADA contributed to nitrogen-containing emissions from bread baking, we baked bread
with and without ADA addition. We used the AMS to measure the composition of PM emissions
during fermentation (i.e., while the bread dough rose) and baking. While we observed OA
emissions during baking, none of our experiments showed the CHN signals with $C_3H_8N^+$,
$C_5H_{12}N^+$, and $C_6H_{14}N^+$. As a result, we cannot conclude that the presence of ADA leads to high
proportions of CHN ions (SI Fig. 7).
Abundant reduced nitrogen was also observed in the particle phase via LC-TOF and LC-
MS/MS measurements. To supplement the online measurements of functionalized aerosol-phase
compounds, especially those containing nitrogen, offline analysis using LC-TOF was employed
for organic compound speciation for each restaurant site with sufficient mass loading, with soft
ionization allowing for the molecular formula-level speciation of observed organic species.
Based on the online AMS data showing differences in OA enhancement (Fig. 2), the samples
were split into three sample groups, the six high-emitting restaurants (Bar/Res 1, Diner 2, Pizza,
Bar/Res 2, Diner 1, Island Cuisine), the lower enhancement near-source cooking samples
(Bakery 1, Bakery 2, Fast Food 1, Fast Food 2, Cafe), and urban samples excluding near-source
cooking samples (i.e., samples taken in different neighborhoods and parks), though this likely
includes cooking-related contributions to the urban background.
Figure 7a shows the ion abundance volatility distribution of the different functionalized
compound classes in the 6 samples with the highest PM concentrations (Fig. 2, see Fig. S10 for
other samples). Compound volatilities were estimated from the generated formulas, assuming all
species were at 300 K (Y. Li et al., 2016) from each sample, and all ion abundances were
normalized by the sample volume for comparison across samples. Figure 7b shows the volatility
distributions of ion abundances from the three sample groups, with the six more enhanced near-





source cooking samples demonstrating high ion abundance consistent with the higher mass
concentrations of PM$_{2.5}$ sampled. The six enhanced cooking samples in Figure 7a show a greater
abundance of I/SVOCs compared to the other two sample groups, suggestive of fresh emissions.
The observed mixtures are highly functionalized, with observed species containing nitrogen,
oxygen, and sulfur, but we note that the LC-TOF employed here has poor ionization efficiencies
for CH and CHS compounds, which are thus not considered for this analysis of functionalized
compounds.

While urban particulate matter has been shown to contain many functionalized species

(Ditto et al., 2018; Ye et al., 2021), recent work has also shown cooking to be a source of
nitrogen and sulfur-containing species, which can be emitted in the gas-phase from foods such as
vegetables (Marcinkowska & Jeleń, 2022) or formed through cooking (Ditto et al., 2022; Takhar
et al., 2019). The urban background samples excluding cooking samples and the five lower
enhanced near-source cooking samples have similar volatility distributions with nitrogen-
containing compounds (Fig. 7b, S11), which suggests a role for cooking emissions in the
background functionalized OA composition in urban areas.

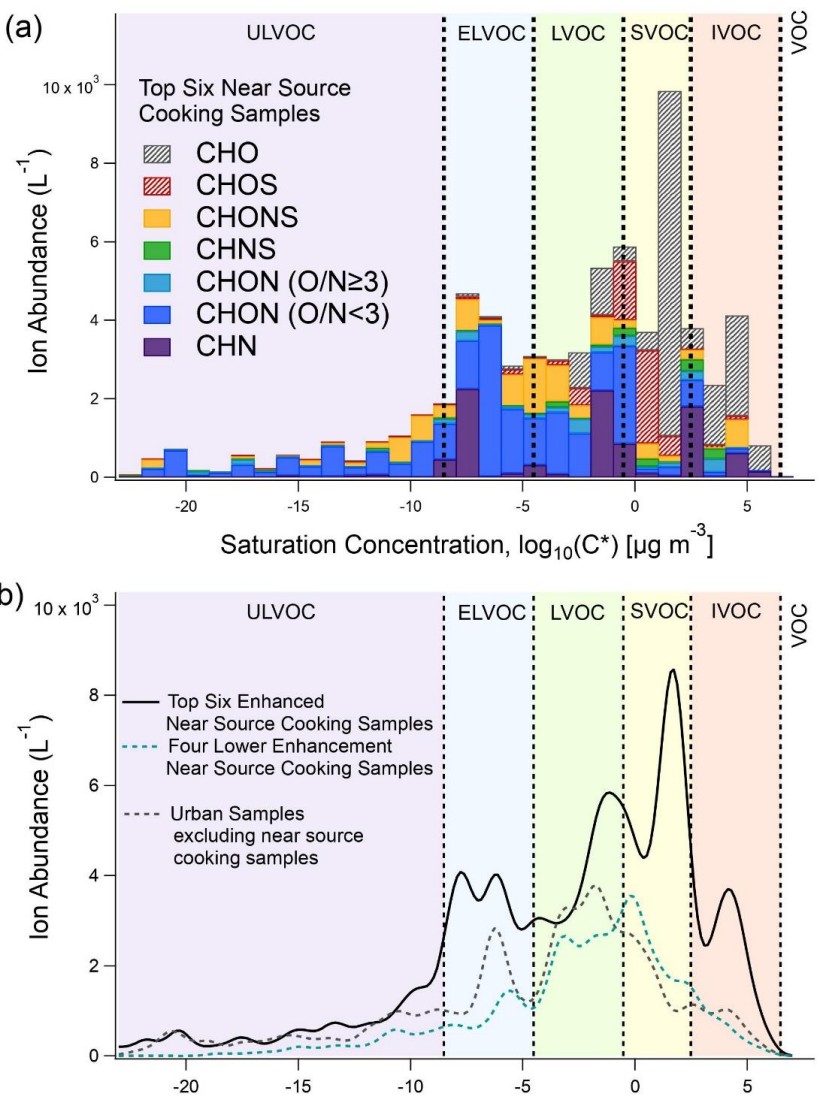

**Figure 7.** Averaged chemical composition of functionalized particle-phase organic compounds
from **(a)** filters collected from the top six near-source cooking samples showing the highest
enhancement in OA from the AMS measurements and **(b)** average ion abundance volatility
distributions for the three sample groups, top six enhanced cooking samples, lower five near-
source cooking samples, and the urban samples excluding near source cooking samples.
Volatility bins were defined for the same reference temperature in **(a)** and **(b)** (i.e., 300 K, as all
samples were collected during summertime).



While all samples contained nitrogen-containing compounds, LC-MS/MS was used on
select samples (Bakery 1, Pizza, background sample 5) from each sample group to compare the
functionalities of observed nitrogen. After compounds observed via LC-TOF (i.e., Fig. 7a)
underwent QC/QA, those compounds were selected for MS/MS analysis in a targeted mode
similar to prior work (Ditto et al., 2020).
Most nitrogen-containing compounds observed had an O/N of less than 3, but other
nitrogen-containing compound classes were present (Fig. 7, S11). Figure 8 shows the observed
nitrogen-containing functional groups for the three samples run on MS/MS, split by O/N ratio
less than 3 or greater than or equal to 3. Here, the Bakery 1 compounds analyzed by MS/MS
were dominated by reduced nitrogen features, with prominent amine and amide functional
groups, especially for compounds with O/N ratios lower than 3, which in itself is indicative of
the presence of reduced nitrogen structural features.



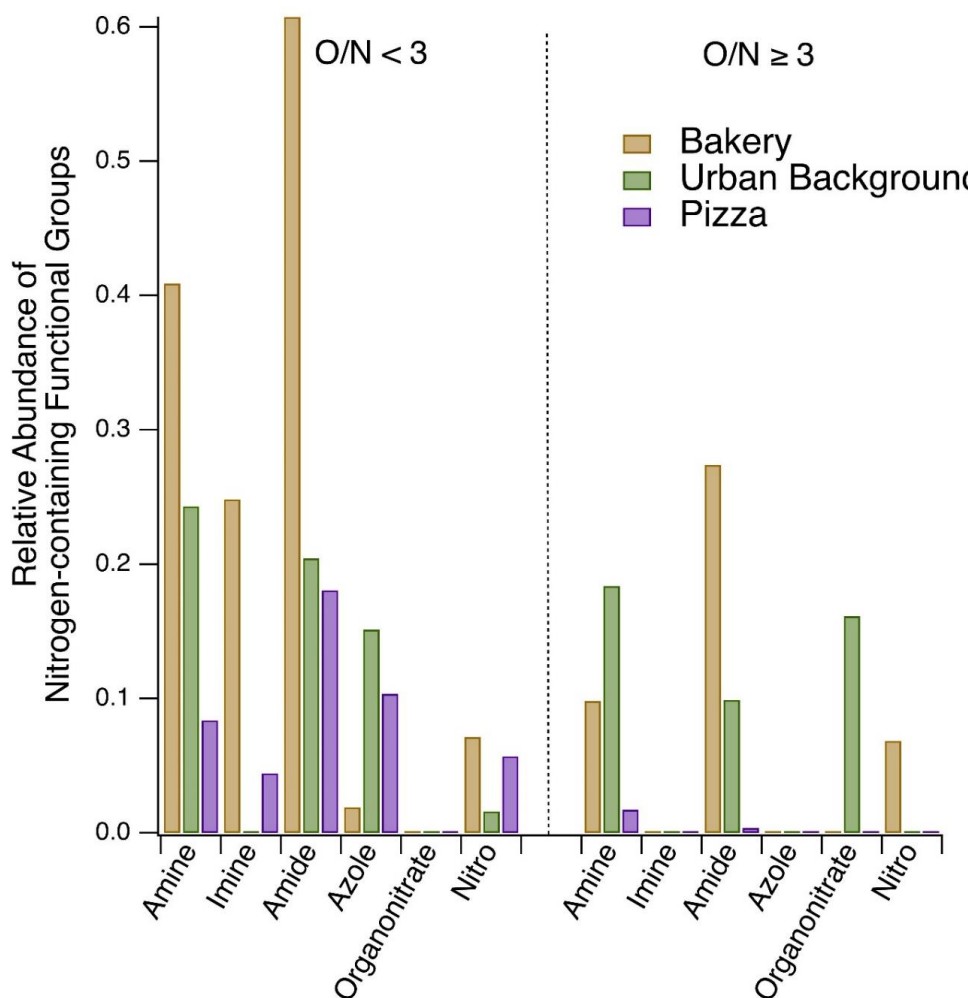


**Figure 8.** The relative abundance of nitrogen-containing functionalities in the Bakery 1,
background sample 5, and Pizza MS/MS compounds are shown, separated by O/N ratio <3 on
the left and ≧ 3 on the right, with prominently reduced nitrogen functionalities in the bakery
sample. See Figure S13 for the complete range of functional groups and structural features
observed in these samples. Enamine, nitrophenol, and nitrile functionalities were also searched
for but were not detected in these three samples.








*3.5. Particle size distributions and UFP enhancements in restaurant plumes*

To expand upon Figure 1's observations of UFPs in an example restaurant plume, we

examined UFP enhancements across the sampled restaurants and the size distribution of those
emissions.

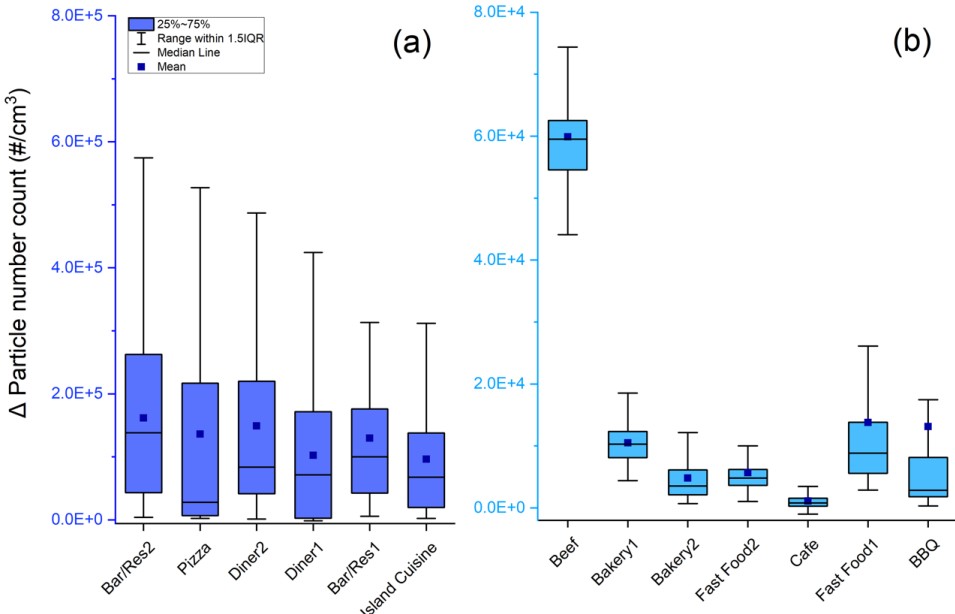


**Figure 9.** Particle number enhancement (ΔPNC) at each restaurant site (with IQR). The sample
names in (a) and (b) are placed in the same order as in Figure 2.
Figure 9 summarizes the particle number concentrations above the background (ΔPNC)

measured by the FMPS and scaled to the CPC. Similar to our ΔOA distribution in Figure 2, there
are notable site-to-site differences in particle number concentrations with the sites breaking down
into the higher and lower-emitting groups (high ΔPNC group mean ΔPNC $> 10^5$ #/cm$^3$, low
ΔPNC group mean ΔPNC $< 10^5$ #/cm$^3$).



All of the high ΔPNC sites were also high ΔOA sites, but most sites do not have a strong
correlation between mean ΔOA and mean ΔPNC (Fig. S8). A moderate positive correlation was
observed in the time series of PNC and OA at Diner 1 ($R^2$ = 0.64), Beef (0.63), Bar/Restaurant 2
(0.60), and Bakery 1 (0.57); most other sites had poor correlations between ΔOA and ΔPNC ($R^2$
< 0.4).  This poor correlation may indicate that the emissions of OA and PNC are decoupled
during cooking so that different activities boost emissions of OA mass versus particle number.
For example, the PNC time series in Figure 1 has several spikes that do not have associated
spikes in OA.
The PNC enhancements are less skewed than the OA enhancements. For ΔPNC, the
mean is always inside the IQR except for the BBQ sample, unlike several sites that had mean
ΔOA > 75th percentile. This implies that PNC emissions are less dominated by intense spikes
than OA emissions. Figure 2 and Figure S4 show that OA concentrations often fell close to the
background between spikes. PNC, on the other hand, was consistently elevated during the
restaurant sampling. One possible explanation is that OA spikes are associated with cooking,
whereas the consistently high PNC is associated with the heating of the cooking surface by either
a natural gas flame or electricity (Amouei Torkmahalleh et al., 2018; Dennekamp et al., 2001;
Wu et al., 2012).



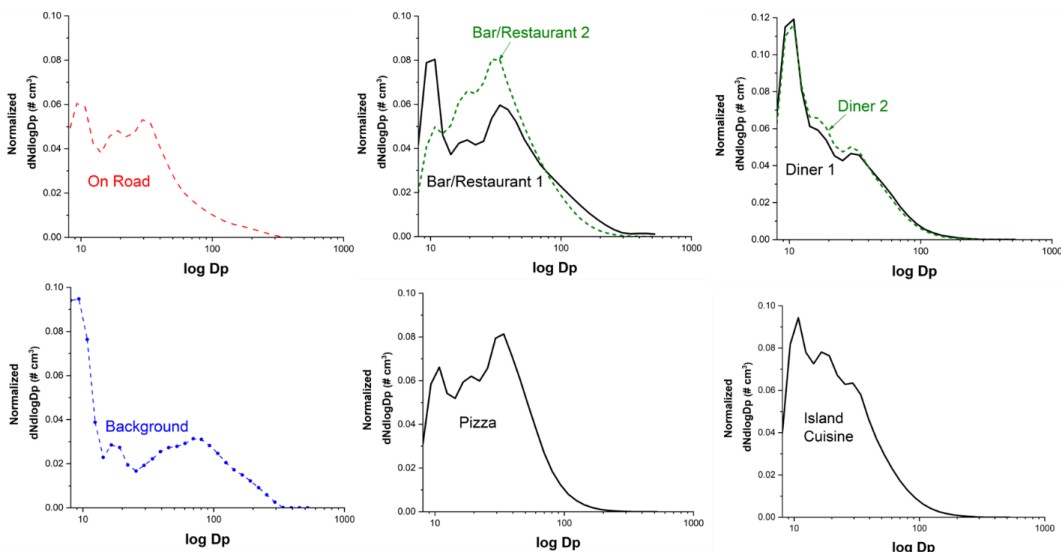


**Figure 10.** Mean particle size distribution comparison of on-road, background, and high ΔPNC
restaurants observed at Bar/Restaurant, Diner, Pizza, and Island cuisine measured from the
FMPS (Fast Mobility Particle Sizer). To fit the size distributions onto the same scale, all are
normalized to the total particle number (dN/dlogDp) of each sampling period over each size bin
and make the sum of all normalized size distributions to be 1.


Figure 10 shows the mean particle size distributions for the "high ΔPNC" restaurants
from Figure 7a and the mean on-road and background particle size distributions from the period
shown in Figure 1. All the restaurants emitted UFPs. The mode particle diameter from all
sampled restaurants was less than 50 nm (Table 1), and the size distributions in Figure 10 clearly
peak in the ultrafine size range. However, there is variability across the restaurants as some sites
had bimodal size distributions, while others are closer to unimodal. For example, Bar/Restaurant
1 has distinct modes at ~10 and 40 nm, whereas Island Cuisine has a single broad mode centered
around 20 nm. There is also variability within sites. For example, Bar/Restaurant 2 has a
unimodal distribution with a mode around 40 nm, and the size distribution differs from the other



sample at the same location, while the two samples at the Diner have nearly identical size
distributions.

In addition to being enhanced in terms of concentrations, the size distributions in the

restaurant plumes are distinct from the average background size distributions, which have a
bimodal distribution with a nucleation mode peak around 10 nm and an accumulation mode peak
around 100 nm. Emissions from nearby vehicles dominate the on-road periods, with a bimodal
size distribution around 10 nm and 20-40 nm, which is similarly observed in previous studies
(Sturm et al., 2003; Wang et al., 2008; X. Yao et al., 2005).

**4. Conclusions and Atmospheric Relevance**

Using mobile measurements across a range of commercial cooking operations in two

cities, our real-world sampling of cooking plumes from restaurants demonstrates substantial
cooking-associated aerosol emissions with variability in the concentrations, chemical
composition, and size distribution of PM and UFP emissions. Reduced nitrogen (N) was
prevalent across all restaurant samples, contributing approximately 15% of the cooking organic
aerosol (OA) mass at the sampled sites, with a diversity of reduced N functional groups
observed. However, a notable finding of this study was the distinct composition of emissions
collected from a commercial bakery, marked by the elevated presence of reduced nitrogen.
Numerous studies have investigated cooking aerosol compositions, demonstrating that different
cooking techniques and ingredients can elevate nitrogen content levels (Ditto et al., 2022;
Masoud et al., 2022; Reyes-Villegas et al., 2018b; Rogge et al., 1991b). Nitrogen found in
cooking emissions has diverse origins, including from the food itself with both natural (e.g.,
protein-rich and plant-based products) (Bak et al., 2019; Han et al., 2020) and anthropogenic



sources (e.g., fertilizers and food additives like nitrates and nitrites) (Dimkpa et al., 2020;
Karwowska & Kononiuk, 2020), as well as from nitrogen dioxide ($NO_2$) and other nitrogen
oxides (NOx) emitted from gas cooking burners (H. Zhao et al., 2021), which are primarily
influenced by the duration of gas cooking and the ambient air quality (Mosqueron et al., 2002).
To further examine potential sources of the nitrogen features identified from the bakery
emissions, we conducted an experiment with the AMS measuring bread baking emissions both
with and without the dough stabilizer azodicarbonamide ($C_2H_4N_4O_2$) as a potential source of N-
containing peaks. While the reduced nitrogen peaks were not observed, this result implies the
challenge in determining specific sources of nitrogen-containing species, particularly in real-
world cooking environments, emphasizing the need for further investigation.

This study also highlights that cooking emissions are substantial contributors to urban

UFPs. Variability between sites was observed, with some sites displaying unimodal and others
displaying bimodal size distributions. However, there are uncertainties in identifying the
characteristics of UFPs from cooking emissions, such as their origin from cooking processes or
natural gas usage, and potential changes in particle size distributions during dilution due to the
evaporation of semi-volatile components. Uncontrolled dilution in this study may have
contributed to differences in UFP concentration and size distribution (Lipsky & Robinson, 2006).

While it is acknowledged that a proportion of cooking emissions may undergo

evaporation during the dilution process, it is improbable that these particles will evaporate
entirely. Previous research conducted by Louvaris et al (Louvaris et al., 2017) investigated meat
charbroiling emissions diluted within a chamber and reported that approximately 80% of the
COA persisted following isothermal dilution at ambient temperature (25 °C) by a factor of 10. In
order to gain a deeper understanding of the factors influencing UFP size distribution from real-



world cooking sources, further investigation is warranted, taking into account aspects such as
restaurant proximity, food type, and order frequency. Consequently, subsequent research can
identify the prevalent molecular features of reduced nitrogen in cooking emissions by setting
constraints on specific parameters, providing a more comprehensive analysis.

Overall, this study underscores the importance of comprehensively understanding

cooking emissions, including their contribution to the $PM_{2.5}$ mass, composition, and exposure
variability across urban areas, in order to develop effective strategies for mitigating their impact
on air quality and human health. Specifically, further research is needed to better understand the
role of reduced nitrogen in atmospheric emissions from cooking activities.


*Data availability.* All data presented in this work can be obtained by directly contacting the
corresponding author at apresto@andrew.cmu.edu upon request.

*Author contributions.* The experimental design was done by AAP and DRG. Data collection was
carried out by AAP and JEM. SK performed the data analysis and compiled the instrumental
data. SK and AAP wrote the paper, with all authors contributing significantly to the
interpretation of the results, discussions, and finalization of the paper.

*Competing interests.* At least one of the (co-)authors is a member of the editorial board of
Atmospheric Chemistry and Physics. The peer-review process was guided by an independent
editor, and the authors also have no other competing interests to declare.



*Acknowledgments.* This research was conducted as part of the Center for Air, Climate, and
Energy Solutions (CACES), which was supported by the Environmental Protection Agency
(assistance agreement number RD83587301) to Carnegie Mellon University. We
acknowledge support from assistance agreement no. RD835871 awarded by the U.S.
Environmental Protection Agency to Yale University. This study has not been formally reviewed
by the EPA. The views expressed in this document are solely those of the authors and do not
necessarily reflect those of the agency. The EPA does not endorse any products or commercial
services mentioned in this publication. DRG and JEM acknowledge financial support from the
U.S. NSF (CBET-2011362). SK and AAP acknowledge funding support from the U.S. NSF
(CBET 1907446)

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

Contributions to Urban Organic Aerosol Using Online Molecular Tracers and Aerosol

Mass Spectrometry Measurements. *Environmental Science & Technology*, *55*(21),

14526–14535. https://doi.org/10.1021/acs.est.1c03280

Huang, X.-F., He, L.-Y., Hu, M., Canagaratna, M. R., Sun, Y., Zhang, Q., Zhu, T., Xue, L.,

Zeng, L.-W., Liu, X.-G., Zhang, Y.-H., Jayne, J. T., Ng, N. L., & Worsnop, D. R. (2010).

Highly time-resolved chemical characterization of atmospheric submicron particles

during 2008 Beijing Olympic Games using an Aerodyne High-Resolution Aerosol Mass

Spectrometer. *Atmospheric Chemistry and Physics*, *10*(18), 8933–8945.

https://doi.org/10.5194/acp-10-8933-2010



Ibald-Mulli, A., Wichmann, H.-E., Kreyling, W., & Peters, A. (2002). Epidemiological Evidence

on Health Effects of Ultrafine Particles. *Journal of Aerosol Medicine*, *15*(2), 189–201.

https://doi.org/10.1089/089426802320282310

Jeong, C.-H., Wang, J. M., & Evans, G. J. (2016). Source Apportionment of Urban Particulate

Matter using Hourly Resolved Trace Metals, Organics, and Inorganic Aerosol

Components. *Atmospheric Chemistry and Physics Discussions*, 1–32.

https://doi.org/10.5194/acp-2016-189

Jung, C.-C., & Su, H.-J. (2020). Chemical and stable isotopic characteristics of PM2.5 emitted

from Chinese cooking. *Environmental Pollution*, *267*, 115577.

https://doi.org/10.1016/j.envpol.2020.115577

Kaltsonoudis, C., Kostenidou, E., Louvaris, E., Psichoudaki, M., Tsiligiannis, E., Florou, K.,

Liangou, A., & Pandis, S. N. (2017). Characterization of fresh and aged organic aerosol

emissions from meat charbroiling. *Atmospheric Chemistry and Physics*, *17*(11), 7143–

7155. https://doi.org/10.5194/acp-17-7143-2017

Karwowska, M., & Kononiuk, A. (2020). Nitrates/Nitrites in Food—Risk for Nitrosative Stress

and Benefits. *Antioxidants*, *9*(3), Article 3. https://doi.org/10.3390/antiox9030241

Keuken, M. P., Roemer, M. G. M., Zandveld, P., Verbeek, R. P., & Velders, G. J. M. (2012).

Trends in primary NO2 and exhaust PM emissions from road traffic for the period 2000–

2020 and implications for air quality and health in the Netherlands. *Atmospheric*

*Environment*, *54*, 313–319. https://doi.org/10.1016/j.atmosenv.2012.02.009

Kiendler-Scharr, A., Zhang, Q., Hohaus, T., Kleist, E., Mensah, A., Mentel, T. F., Spindler, C.,

Uerlings, R., Tillmann, R., & Wildt, J. (2009). Aerosol Mass Spectrometric Features of

Biogenic SOA: Observations from a Plant Chamber and in Rural Atmospheric



Environments. *Environmental Science & Technology*, *43*(21), 8166–8172.

https://doi.org/10.1021/es901420b

Klompmaker, J. O., Montagne, D. R., Meliefste, K., Hoek, G., & Brunekreef, B. (2015). Spatial

variation of ultrafine particles and black carbon in two cities: Results from a short-term

measurement campaign. *Science of The Total Environment*, *508*, 266–275.

https://doi.org/10.1016/j.scitotenv.2014.11.088

Kostenidou, E., Lee, B.-H., Engelhart, G. J., Pierce, J. R., & Pandis, S. N. (2009). Mass Spectra

Deconvolution of Low, Medium, and High Volatility Biogenic Secondary Organic

Aerosol. *Environmental Science & Technology*, *43*(13), 4884–4889.

https://doi.org/10.1021/es803676g

Kwon, H.-S., Ryu, M. H., & Carlsten, C. (2020). Ultrafine particles: Unique physicochemical

properties relevant to health and disease. *Experimental & Molecular Medicine*, *52*(3),

Article 3. https://doi.org/10.1038/s12276-020-0405-1

Lalchandani, V., Kumar, V., Tobler, A., M. Thamban, N., Mishra, S., Slowik, J. G., Bhattu, D.,

Rai, P., Satish, R., Ganguly, D., Tiwari, S., Rastogi, N., Tiwari, S., Močnik, G., Prévôt,

845        A. S. H., & Tripathi, S. N. (2021). Real-time characterization and source apportionment

of fine particulate matter in the Delhi megacity area during late winter. *Science of The*

*Total Environment*, *770*, 145324. https://doi.org/10.1016/j.scitotenv.2021.145324

Lanz, V. A., Alfarra, M. R., Baltensperger, U., Buchmann, B., Hueglin, C., & Prévôt, A. S. H.

(2007). Source apportionment of submicron organic aerosols at an urban site by factor

analytical modelling of aerosol mass spectra. *Atmospheric Chemistry and Physics*, *7*(6),

1503–1522. https://doi.org/10.5194/acp-7-1503-2007



Lee, B. P., Li, Y. J., Yu, J. Z., Louie, P. K. K., & Chan, C. K. (2015). Characteristics of
submicron particulate matter at the urban roadside in downtown Hong Kong—Overview
of 4 months of continuous high-resolution aerosol mass spectrometer measurements.
*Journal of Geophysical Research: Atmospheres*, *120*(14), 7040–7058.
https://doi.org/10.1002/2015JD023311
Lenschow, P., Abraham, H.-J., Kutzner, K., Lutz, M., Preuß, J.-D., & Reichenbächer, W. (2001).
Some ideas about the sources of PM10. *Atmospheric Environment*, *35*, S23–S33.
https://doi.org/10.1016/S1352-2310(01)00122-4
Li, Y., Pöschl, U., & Shiraiwa, M. (2016). Molecular corridors and parameterizations of
volatility in the chemical evolution of organic aerosols. *Atmospheric Chemistry and*
*Physics*, *16*(5), 3327–3344. https://doi.org/10.5194/acp-16-3327-2016
Li, Z., Fung, J. C. H., & Lau, A. K. H. (2018). High spatiotemporal characterization of on-road
PM2.5 concentrations in high-density urban areas using mobile monitoring. *Building and*
*Environment*, *143*, 196–205. https://doi.org/10.1016/j.buildenv.2018.07.014
Liu, T., Li, Z., Chan, M., & Chan, C. K. (2017). Formation of secondary organic aerosols from
gas-phase emissions of heated cooking oils. *Atmospheric Chemistry and Physics*, *17*(12),
7333–7344. https://doi.org/10.5194/acp-17-7333-2017
Liu, T., Wang, Z., Wang, X., & Chan, C. K. (2018). Primary and secondary organic aerosol from
heated cooking oil emissions. *Atmospheric Chemistry and Physics*, *18*(15), 11363–11374.
https://doi.org/10.5194/acp-18-11363-2018
Louie, P. K. K., Chow, J. C., Chen, L.-W. A., Watson, J. G., Leung, G., & Sin, D. W. M. (2005).
PM2.5 chemical composition in Hong Kong: Urban and regional variations. *Science of*
*The Total Environment*, *338*(3), 267–281. https://doi.org/10.1016/j.scitotenv.2004.07.021



Louvaris, E. E., Karnezi, E., Kostenidou, E., Kaltsonoudis, C., & Pandis, S. N. (2017).
Estimation of the volatility distribution of organic aerosol combining thermodenuder and
isothermal dilution measurements. *Atmospheric Measurement Techniques*, *10*(10), 3909–
3918. https://doi.org/10.5194/amt-10-3909-2017
Marcinkowska, M. A., & Jeleń, H. H. (2022). Role of Sulfur Compounds in Vegetable and
Mushroom Aroma. *Molecules*, *27*(18), Article 18.
https://doi.org/10.3390/molecules27186116
Masoud, C. G., Li, Y., Wang, D. S., Katz, E. F., DeCarlo, P. F., Farmer, D. K., Vance, M. E.,
Shiraiwa, M., & Hildebrandt Ruiz, L. (2022). Molecular composition and gas-particle
partitioning of indoor cooking aerosol: Insights from a FIGAERO-CIMS and kinetic
aerosol modeling. *Aerosol Science and Technology*, *0*(0), 1–18.
https://doi.org/10.1080/02786826.2022.2133593
Milic, A., Miljevic, B., Alroe, J., Mallet, M., Canonaco, F., Prevot, A. S. H., & Ristovski, Z. D.
(2016). The ambient aerosol characterization during the prescribed bushfire season in
Brisbane 2013. *Science of The Total Environment*, *560–561*, 225–232.
https://doi.org/10.1016/j.scitotenv.2016.04.036
Mohr, C., Huffman, J. A., Cubison, M. J., Aiken, A. C., Docherty, K. S., Kimmel, J. R., Ulbrich,
I. M., Hannigan, M., & Jimenez, J. L. (2009). Characterization of Primary Organic
Aerosol Emissions from Meat Cooking, Trash Burning, and Motor Vehicles with High-
Resolution Aerosol Mass Spectrometry and Comparison with Ambient and Chamber
Observations. *Environmental Science & Technology*, *43*(7), 2443–2449.
https://doi.org/10.1021/es8011518



Mohr, C., Richter, R., DeCarlo, P. F., Prévôt, A. S. H., & Baltensperger, U. (2011). Spatial

variation of chemical composition and sources of submicron aerosol in Zurich during

wintertime using mobile aerosol mass spectrometer data. *Atmospheric Chemistry and*

*Physics*, *11*(15), 7465–7482. https://doi.org/10.5194/acp-11-7465-2011

Mosqueron, L., Momas, I., & Moullec, Y. L. (2002). Personal exposure of Paris office workers

to nitrogen dioxide and fine particles. *Occupational and Environmental Medicine*, *59*(8),

550–555. https://doi.org/10.1136/oem.59.8.550

N. Pandis, S., Skyllakou, K., Florou, K., Kostenidou, E., Kaltsonoudis, C., Hasa, E., & A. Presto,

905          A. (2016). Urban particulate matter pollution: A tale of five cities. *Faraday Discussions*,

*189*(0), 277–290. https://doi.org/10.1039/C5FD00212E

Omelekhina, Y., Eriksson, A., Canonaco, F., H. Prevot, A. S., Nilsson, P., Isaxon, C., Pagels, J.,

& Wierzbicka, A. (2020). Cooking and electronic cigarettes leading to large differences

between indoor and outdoor particle composition and concentration measured by aerosol

mass spectrometry. *Environmental Science: Processes & Impacts*, *22*(6), 1382–1396.

https://doi.org/10.1039/D0EM00061B

Raatikainen, T., Vaattovaara, P., Tiitta, P., Miettinen, P., Rautiainen, J., Ehn, M., Kulmala, M.,

Laaksonen, A., & Worsnop, D. R. (2010). Physicochemical properties and origin of

organic groups detected in boreal forest using an aerosol mass spectrometer. *Atmospheric*

*Chemistry and Physics*, *10*(4), 2063–2077. https://doi.org/10.5194/acp-10-2063-2010

Renzi, M., Marchetti, S., de' Donato, F., Pappagallo, M., Scortichini, M., Davoli, M., Frova, L.,

Michelozzi, P., & Stafoggia, M. (2021). Acute Effects of Particulate Matter on All-Cause

Mortality in Urban, Rural, and Suburban Areas, Italy. *International Journal of*



*Environmental Research and Public Health*, *18*(24), Article 24.
https://doi.org/10.3390/ijerph182412895
Reyes-Villegas, E., Bannan, T., Le Breton, M., Mehra, A., Priestley, M., Percival, C., Coe, H., &
Allan, J. D. (2018a). Online Chemical Characterization of Food-Cooking Organic
Aerosols: Implications for Source Apportionment. *Environmental Science & Technology*,
*52*(9), 5308–5318. https://doi.org/10.1021/acs.est.7b06278
Reyes-Villegas, E., Bannan, T., Le Breton, M., Mehra, A., Priestley, M., Percival, C., Coe, H., &
Allan, J. D. (2018b). Online Chemical Characterization of Food-Cooking Organic
Aerosols: Implications for Source Apportionment. *Environmental Science & Technology*,
*52*(9), 5308–5318. https://doi.org/10.1021/acs.est.7b06278
Rogge, W. F., Hildemann, L. M., Mazurek, M. A., Cass, G. R., & Simoneit, B. R. T. (1991a).
Sources of fine organic aerosol. 1. Charbroilers and meat cooking operations.
*Environmental Science & Technology*, *25*(6), 1112–1125.
https://doi.org/10.1021/es00018a015
Rogge, W. F., Hildemann, L. M., Mazurek, M. A., Cass, G. R., & Simoneit, B. R. T. (1991b).
Sources of fine organic aerosol. 1. Charbroilers and meat cooking operations.
*Environmental Science & Technology*, *25*(6), 1112–1125.
https://doi.org/10.1021/es00018a015
Rose Eilenberg, S., Subramanian, R., Malings, C., Hauryliuk, A., Presto, A. A., & Robinson, A.
L. (2020). Using a network of lower-cost monitors to identify the influence of modifiable
factors driving spatial patterns in fine particulate matter concentrations in an urban
environment. *Journal of Exposure Science & Environmental Epidemiology*, *30*(6), Article
6. https://doi.org/10.1038/s41370-020-0255-x



Ruggeri, G., & Takahama, S. (2016). Technical Note: Development of chemoinformatic tools to

enumerate functional groups in molecules for organic aerosol characterization.

*Atmospheric Chemistry and Physics*, *16*(7), 4401–4422. https://doi.org/10.5194/acp-16-

4401-2016

Saha, P. K., Sengupta, S., Adams, P., Robinson, A. L., & Presto, A. A. (2020). Spatial

Correlation of Ultrafine Particle Number and Fine Particle Mass at Urban Scales:

Implications for Health Assessment. *Environmental Science & Technology*, *54*(15),

9295–9304. https://doi.org/10.1021/acs.est.0c02763

Saha, P. K., Zimmerman, N., Malings, C., Hauryliuk, A., Li, Z., Snell, L., Subramanian, R.,

Lipsky, E., Apte, J. S., Robinson, A. L., & Presto, A. A. (2019). Quantifying high-

resolution spatial variations and local source impacts of urban ultrafine particle

concentrations. *Science of The Total Environment*, *655*, 473–481.

https://doi.org/10.1016/j.scitotenv.2018.11.197

Schauer, J. J., Kleeman, M. J., Cass, G. R., & Simoneit, B. R. T. (2002). Measurement of

Emissions from Air Pollution Sources. 4. C1−C27 Organic Compounds from Cooking

with Seed Oils. *Environmental Science & Technology*, *36*(4), 567–575.

https://doi.org/10.1021/es002053m

Schauer, J. J., Rogge, W. F., Hildemann, L. M., Mazurek, M. A., Cass, G. R., & Simoneit, B. R.

960        T. (1996). Source apportionment of airborne particulate matter using organic compounds

as tracers. *Atmospheric Environment*, *30*(22), 3837–3855. https://doi.org/10.1016/1352-

2310(96)00085-4

Schraufnagel, D. E. (2020). The health effects of ultrafine particles. *Experimental & Molecular*

*Medicine*, *52*(3), Article 3. https://doi.org/10.1038/s12276-020-0403-3





Shah, R. U., Robinson, E. S., Gu, P., Robinson, A. L., Apte, J. S., & Presto, A. A. (2018). High-

spatial-resolution mapping and source apportionment of aerosol composition in Oakland,

California, using mobile aerosol mass spectrometry. *Atmospheric Chemistry and Physics*,

*18*(22), 16325–16344. https://doi.org/10.5194/acp-18-16325-2018

Song, R., Presto, A. A., Saha, P., Zimmerman, N., Ellis, A., & Subramanian, R. (2021a). Spatial

variations in urban air pollution: Impacts of diesel bus traffic and restaurant cooking at

small scales. *Air Quality, Atmosphere & Health*. https://doi.org/10.1007/s11869-021-

01078-8

Song, R., Presto, A. A., Saha, P., Zimmerman, N., Ellis, A., & Subramanian, R. (2021b). Spatial

variations in urban air pollution: Impacts of diesel bus traffic and restaurant cooking at

small scales. *Air Quality, Atmosphere & Health*, *14*(12), 2059–2072.

https://doi.org/10.1007/s11869-021-01078-8

Sturm, P. J., Baltensperger, U., Bacher, M., Lechner, B., Hausberger, S., Heiden, B., Imhof, D.,

Weingartner, E., Prevot, A. S. H., Kurtenbach, R., & Wiesen, P. (2003). Roadside

measurements of particulate matter size distribution. *Atmospheric Environment*, *37*(37),

5273–5281. https://doi.org/10.1016/j.atmosenv.2003.05.006

Sun, Y. L., Wang, Z. F., Fu, P. Q., Yang, T., Jiang, Q., Dong, H. B., Li, J., & Jia, J. J. (2013).

Aerosol composition, sources and processes during wintertime in Beijing, China.

*Atmospheric Chemistry and Physics*, *13*(9), 4577–4592. https://doi.org/10.5194/acp-13-

4577-2013

Sun, Y. L., Zhang, Q., Schwab, J. J., Yang, T., Ng, N. L., & Demerjian, K. L. (2012). Factor

analysis of combined organic and inorganic aerosol mass spectra from high resolution



aerosol mass spectrometer measurements. *Atmospheric Chemistry and Physics*, *12*(18),
8537–8551. https://doi.org/10.5194/acp-12-8537-2012
Takhar, M., Stroud, C. A., & Chan, A. W. H. (2019). Volatility Distribution and Evaporation
Rates of Organic Aerosol from Cooking Oils and their Evolution upon Heterogeneous
Oxidation. *ACS Earth and Space Chemistry*, *3*(9), 1717–1728.
https://doi.org/10.1021/acsearthspacechem.9b00110
Tan, Y., Dallmann, T. R., Robinson, A. L., & Presto, A. A. (2016). Application of plume
analysis to build land use regression models from mobile sampling to improve model
transferability. *Atmospheric Environment*, *134*, 51–60.
https://doi.org/10.1016/j.atmosenv.2016.03.032
Torkmahalleh, M. A., Goldasteh, I., Zhao, Y., Udochu, N. M., Rossner, A., Hopke, P. K., &
Ferro, A. R. (2012). PM2.5 and ultrafine particles emitted during heating of commercial
cooking oils. *Indoor Air*, *22*(6), 483–491. https://doi.org/10.1111/j.1600-
0668.2012.00783.x
Wallace, L. A., Emmerich, S. J., & Howard-Reed, C. (2004). Source Strengths of Ultrafine and
Fine Particles Due to Cooking with a Gas Stove. *Environmental Science & Technology*,
*38*(8), 2304–2311. https://doi.org/10.1021/es0306260
Wan, M.-P., Wu, C.-L., Sze To, G.-N., Chan, T.-C., & Chao, C. Y. H. (2011). Ultrafine particles,
and PM2.5 generated from cooking in homes. *Atmospheric Environment*, *45*(34), 6141–
6148. https://doi.org/10.1016/j.atmosenv.2011.08.036
Wang, Y., Bechle, M. J., Kim, S.-Y., Adams, P. J., Pandis, S. N., Pope, C. A., Robinson, A. L.,
Sheppard, L., Szpiro, A. A., & Marshall, J. D. (2020). Spatial decomposition analysis of



NO2 and PM2.5 air pollution in the United States. *Atmospheric Environment*, *241*,

117470. https://doi.org/10.1016/j.atmosenv.2020.117470

Wang, Y., Zhu, Y., Salinas, R., Ramirez, D., Karnae, S., & John, K. (2008). Roadside

Measurements of Ultrafine Particles at a Busy Urban Intersection. *Journal of the Air &*

*Waste Management Association*, *58*(11), 1449–1457. https://doi.org/10.3155/1047-

3289.58.11.1449

Wu, C. L., Chao, C. Y. H., Sze-To, G. N., Wan, M. P., & Chan, T. C. (2012). Ultrafine Particle

Emissions from Cigarette Smouldering, Incense Burning, Vacuum Cleaner Motor

Operation and Cooking. *Indoor and Built Environment*, *21*(6), 782–796.

https://doi.org/10.1177/1420326X11421356

Yao, D., Lyu, X., Lu, H., Zeng, L., Liu, T., Chan, C. K., & Guo, H. (2021). Characteristics,

sources and evolution processes of atmospheric organic aerosols at a roadside site in

Hong Kong. *Atmospheric Environment*, *252*, 118298.

https://doi.org/10.1016/j.atmosenv.2021.118298

Yao, X., Lau, N. T., Fang, M., & Chan, C. K. (2005). Real-Time Observation of the

Transformation of Ultrafine Atmospheric Particle Modes. *Aerosol Science and*

*Technology*, *39*(9), 831–841. https://doi.org/10.1080/02786820500295248

Ye, C., Yuan, B., Lin, Y., Wang, Z., Hu, W., Li, T., Chen, W., Wu, C., Wang, C., Huang, S., Qi,

1027        J., Wang, B., Wang, C., Song, W., Wang, X., Zheng, E., Krechmer, J. E., Ye, P., Zhang,

Z., … Shao, M. (2021). Chemical characterization of oxygenated organic compounds in

the gas phase and particle phase using iodide CIMS with FIGAERO in urban air.

*Atmospheric Chemistry and Physics*, *21*(11), 8455–8478. https://doi.org/10.5194/acp-21-

8455-2021



Zhang, Y., Tang, L., Yu, H., Wang, Z., Sun, Y., Qin, W., Chen, W., Chen, C., Ding, A., Wu, J., Ge, S., Chen, C., & Zhou, H. (2015). Chemical composition, sources and evolution processes of aerosol at an urban site in Yangtze River Delta, China during wintertime. *Atmospheric Environment*, *123*, 339–349. https://doi.org/10.1016/j.atmosenv.2015.08.017

Zhang, Z., Zhu, W., Hu, M., Wang, H., Chen, Z., Shen, R., Yu, Y., Tan, R., & Guo, S. (2021). Secondary Organic Aerosol from Typical Chinese Domestic Cooking Emissions. *Environmental Science & Technology Letters*, *8*(1), 24–31. https://doi.org/10.1021/acs.estlett.0c00754

Zhao, H., Chan, W. R., Cohn, S., Delp, W. W., Walker, I. S., & Singer, B. C. (2021). Indoor air quality in new and renovated low-income apartments with mechanical ventilation and natural gas cooking in California. *Indoor Air*, *31*(3), 717–729. https://doi.org/10.1111/ina.12764

Zhao, Y., Hu, M., Slanina, S., & Zhang, Y. (2007a). Chemical Compositions of Fine Particulate Organic Matter Emitted from Chinese Cooking. *Environmental Science & Technology*, *41*(1), 99–105. https://doi.org/10.1021/es0614518

Zhao, Y., Hu, M., Slanina, S., & Zhang, Y. (2007b). The molecular distribution of fine particulate organic matter emitted from Western-style fast food cooking. *Atmospheric Environment*, *41*(37), 8163–8171. https://doi.org/10.1016/j.atmosenv.2007.06.029

Zhu, Q., Huang, X.-F., Cao, L.-M., Wei, L.-T., Zhang, B., He, L.-Y., Elser, M., Canonaco, F., Slowik, J. G., Bozzetti, C., El-Haddad, I., & Prévôt, A. S. H. (2018). Improved source apportionment of organic aerosols in complex urban air pollution using the multilinear



engine (ME-2). *Atmospheric Measurement Techniques*, *11*(2), 1049–1060.

https://doi.org/10.5194/amt-11-1049-2018

1056