# Peer review of "Real-world observations of ultrafine particles and reduced nitrogen in"

_EGUsphere, 2023_

## Author Comment (AC1)

My main concern is that the sample inlet (line 107) was a few meters from the exhaust vent. This suggests that the measurements will be mixed with ambient concentrations and also affected by wind speed and direction.

We acknowledge the impact of dilution effects. Lines 112-116 highlight the potential variability from ambient background air mixing. Field campaigns inherently face unpredictability compared to controlled lab settings. Despite these challenges, they reflect real-world conditions. We delve deeper into these effects in Section 4, "Conclusion and Atmospheric Relevance."

Another concern is the fact that the authors measured in background, on-road and cooking environments at different times of the day and analysed the differences between sites, also calculating Δconcentrations (ΔOA and ΔBC) based on the background concentrations. Please, explain why you can do this analysis.

Using our mobile laboratory, we measured emissions from background, on-road, and cooking environments. While our initial plan didn't include assessing these diverse sites, driving the lab van to the cooking sample location inadvertently led us past the background site and several roads. This unintentional movement provided the opportunity to collect data from these extra sites.

Figure 1 graphically presents the differences in emission concentrations across these locations, highlighting the distinct characteristics of cooking emissions in contrast to the incidental background and road sites.

Specific comments.

Line 44. Depending on the location, biomass burning is also a main OA source.

We acknowledge biomass burning as a major source of Organic Aerosols (OA). Yet, our study specifically focuses on urban air pollution sources. We understand there might have been some ambiguity in our manuscript, leading to misconceptions.

To clarify, we've revised statements, notably in line 44, to emphasize our concentration on urban air quality in the United States.

Line 59. The autors mention PM2.5 cooking has increase but the data used to make their point (line 56) states that their concentrations went from 2.4 µg/m3 to 1.2 µg/m3 56 between 1982 and 2010.

We have emphasized the increasing percentage of $PM_{2.5}$ attributed to cooking. This may seem contradictory given the overall decrease in $PM_{2.5}$ levels and reduced traffic emissions. However, it's

important to clarify that while the absolute amount of PM$_{2.5}$ from cooking has decreased, its relative contribution to the total PM$_{2.5}$ has increased when compared to studies from 1982 and 2010. In essence, over time, cooking's proportionate share of PM$_{2.5}$ has grown, even if its absolute contribution has lessened.

Line 164. Are you sure that the AMS spectra with nitrogen-containing compounds are from the cooking exhaust?

We adhered to methodologies endorsed by the AMS user group meeting to ensure the utmost precision in our data. To validate the presence of N-containing peaks in our mass spectrum, we examined the following steps:

First, one potential concern is that the formation of refractory components on the vaporizer surface can lead to conditioning that potentially affects the vaporizer interaction (Allan et al., 2003). However, after consultation with Aerodyne, it was confirmed that our signals were not due to surface ionization. Notably, peak shapes and peak widths of ions like Na+ and K+ were similar to Ar+, as seen in Figure S9. This similarity suggests consistent peak shapes without anomalies. If the signals from those metal elements show distinct lumpy or broad signals, it indicates that the ions are coming from the surface of the vaporizer and follow a different path to get extracted into the MS, leading to different peak shapes.

Second, Figure S8 demonstrates the peak fitting of N-containing fragments, providing their genuine existence. Further assurance comes from the absence of signals when the chopper was closed, ruling out internal instrumental errors.

After thorough checks, we can assertively confirm our data's accuracy and eliminate the possibility of instrumental anomalies influencing the observed nitrogen contributions.

*References:*

*Allan, J. D., Alfarra, M. R., Bower, K. N., Williams, P. I., Gallagher, M. W., Jimenez, J. L., ... & Worsnop, D. R. (2003). Quantitative sampling using an Aerodyne aerosol mass spectrometer 2. Measurements of fine particulate chemical composition in two UK cities. Journal of Geophysical Research: Atmospheres, 108(D3).*

Looking at the supplement, it seems that the sampling inlet is too low and perhaps to far from the cooking exhaust (figure S1.). The measurements performed by the instruments might be mixed with the ambient OA concentrations.

We maintained a consistent sampling inlet position across all sites using a single mobile laboratory. However, this consistency might lead to varied dilution rates with the background atmosphere at each site. Elements like restaurant vent placements and atmospheric conditions such as wind speed and direction can affect outcomes depending on the sampling day and location. This could result in a mix of ambient Organic Aerosols (OA) with delta OA and Black Carbon (BC) concentrations. Figure 1

adjusts for this by removing background pollutants from the initial data. We acknowledge the complexities introduced by these factors and have sought to thoroughly address them in our study.

**Line 181. The sampling methodology should be moved to methods section.**

The methods section of our paper provides an in-depth look at our sampling approach. The details pertaining to Figure 1 specifically shed light on the timing of our background, on-road, and cooking emissions measurements. Through a succinct summary of our method in this context, we strive to improve the reader's understanding of the figure in relation to our broader methodology.

**Line 186. These variations in concentrations could also be related to the different times of the day that measurements were taken. i.e. from 3pm to 6pm.**

Figure 1 reflects variations based on two main factors: sampling location and time. Notably, these dynamics only stabilized once we reached the restaurant emission source. For context, the sample from Bar/Restaurant 2 was taken on August 14, 2019, with sampling commencing at 6:20 pm and concluding at 7:04 pm. As depicted in Figure 1, as we neared the restaurant around 6:20 pm, concentrations of $CO$, $CO_2$, Organics, Black Carbon, and Particulate number markedly rose. This spike isn't easily attributed to just time variations, as it's not a slow increase. It's a pronounced surge, likely due to closeness to the emission source.

**Line 190. I would not just call it "likey" when you are measuring during the on-road sampling period, vehicles are the main pollution source.**

In line 195, we've chosen to use "significantly" instead of "likely" to better emphasize vehicles as the predominant pollution source on roads.

**Line 194. I´m not sure about the calculation of a ΔOA and ΔBC**

To accurately assess air quality, background pollutant levels are typically subtracted to focus on specific source effects. This 'background' originates from natural events and distant or widespread human-made sources. Subtracting these levels accentuates the impact of the primary source under study. This method effectively isolates contributions from natural and distant sources, providing a more distinct view of the specific source's pollution impact. In our work, we derived ΔOA and ΔBC by subtracting background levels (based on the 5th percentile of daily data) from the measured OA or BC levels. This approach refines our insight by considering inherent and distant pollutant influences.

Line 216. This could also be related to a change in wind direction or wind speed.

We agree that numerous factors can affect spikes in OA concentrations. But when our mobile laboratory was proximate to restaurant emission sources, cooking emissions or other outputs from these establishments likely played the dominant role in these peaks. Although wind changes could cause mixing with the background atmosphere, potentially diluting OA levels, the notable surge in concentration is evident. As previously emphasized, this sharp rise isn't easily ascribed only to day-to-day fluctuations. The data reveals a distinct and immediate spike, pointing more towards the nearby emission source than to other external variables.

Line 235. The authors should explain why they are calculating the OA/BC or maybe at the end of this analysis to give an explanation about the meaning of these values.

In our study, the OA/BC ratio was instrumental in identifying the sources and trajectories of emissions. This ratio can vary based on factors like food type, cooking techniques, and chosen cooking oil or fat. Organic emissions mainly stem from the heating and pyrolysis of oils, fats, and food, while black carbon emissions predominantly come from burning biomass fuels and coal. Evaluating the OA/BC ratio sheds light on the balance between primary emissions and incomplete combustions, the latter signaled by black carbon levels. This ratio also highlights organic aerosol contributions specific to cooking. This comprehensive approach greatly enhances our understanding of the diverse emission sources.

Line 244. In order to mention that CO may come from food cooking rather than from fuel combustion the authors should know about the cooking practices and fuels. This is an important information.

Line 363. Those peaks are also related to combustion, for example the hydrocarbon OA factor from vehicle emissions. That is why it is important to know the fuel type used in the restaurants.

In our research, we drew informed assumptions from our observations and a thorough review of relevant literature. We observed that most restaurants we studied seemed to use typical stoves, neither electric nor wood-fired. However, given the inherent variables of real-world field studies compared to controlled lab experiments, we can't definitively specify the fuel type or combustion method each restaurant used, both of which profoundly affect emission compositions. Consequently, our methodology involved making inferences from observable factors during sampling and insights from prior studies.

---

## Author Comment (AC2)

1. The title alludes to observations of ultrafine particles while the manuscript mainly focuses on OA AMS measurements and not UFP. The title should be changed to reflect the content of the paper.

To more accurately convey our study's main emphasis, we've revised the manuscript title from "Real-world observations of ultrafine particles and reduced nitrogen in commercial cooking organic aerosol emissions" to "Real-world observations of reduced nitrogen and ultrafine particles in commercial cooking organic aerosol emissions".

This updated title encapsulates the core facets of our research, accentuating our principal concentration on OA resulting from cooking emissions.

2. The importance of nitrogen related species (CHN family) in OA spectra, was measured to be on average 10-15% of OA, with the exception of bakery data (up to 50%). The high contribution of CHN family is not well addressed through the manuscript without concluding to robust results.

To validate the presence of N-containing peaks in our mass spectrum, we examined the following steps:

First, one potential concern is that the formation of refractory components on the vaporizer surface can lead to conditioning that potentially affects the vaporizer interaction (Allan et al., 2003). However, after consultation with Aerodyne, it was confirmed that our signals were not due to surface ionization. Notably, peak shapes and peak widths of ions like Na+ and K+ were similar to Ar+, as seen in Figure S9. This similarity suggests consistent peak shapes without anomalies. If the signals from those metal elements show distinct lumpy or broad signals, it indicates that the ions are coming from the surface of the vaporizer and follow a different path to get extracted into the MS, leading to different peak shapes.

Second, Figure S8 demonstrates the peak fitting of N-containing fragments, providing their genuine existence. Further assurance comes from the absence of signals when the chopper was closed, ruling out internal instrumental errors.

In conclusion, the nitrogen-containing fragments are not results of surface ionization.

*References:*

*Allan, J. D., Alfarra, M. R., Bower, K. N., Williams, P. I., Gallagher, M. W., Jimenez, J. L., ... & Worsnop, D. R. (2003). Quantitative sampling using an Aerodyne aerosol mass spectrometer 2. Measurements of fine particulate chemical composition in two UK cities. Journal of Geophysical Research: Atmospheres, 108(D3).*

3. Bakery 1 and bakery 2 data refer to the same bakery (during different days) but without knowing what practices are used by the bakery personnel, which makes the data problematic. The data could be irrelevant to cooking/ bread making and may relate even to cleaning products or chimney residues. Consequently, bakery data should be treated differently and not characterized as COA – this is supported by the low R and high theta angle once comparing the bakery OA mass spectra to those either from literature COA or other locations presented here. The lack of cooking conditions/information applies to all data, and this should be taken into account.

In our study, 'cooking emissions' encompass emissions from the entire cooking process in a restaurant, from ingredient prep to post-cooking cleanup. Under this definition, the Bakery site's baking activities qualify as cooking emissions.

We recognize, as mentioned from lines 112-116, the potential blending of ambient air with cooking emissions and understand the variability it might introduce. While we address the uncertainties in pinpointing emission origins and potential dilution effects at line 607, it's crucial to note that such challenges are typical in field campaigns, and our analytical approach accommodates the complexities of real-world research. In addition, we elaborate on these effects and uncertainties in Section 4, "Conclusion and Atmospheric Relevance."

4. More discussion is needed regarding COA/HOA similarities/ differences. The current work can aid the distinguishing of the two sources.

Both Cooking Organic Aerosol (COA) and Hydrocarbon-like Organic Aerosol (HOA) are typically classified as factors of Primary Organic Aerosol (POA). Consequently, they may demonstrate similar traits, particularly in their high contributions from hydrocarbon fragments. Furthermore, there exists a possibility that these elements might mix slightly within each respective factor, as noted by Chen et al., 2022. The HOA factor, as per existing research, often exhibits a pronounced diel pattern, characterized by distinct peaks during morning and evening rush hours. Additionally, the HOA factor profile typically shares similarities across diverse sites (Crippa et al., 2014). Turning to COA, previous AMS studies have suggested that $m/z$ 41 ($C_3H_5^+$), $m/z$ 43 ($C_2H_3O^+$ and $C_3H_7^+$), and $m/z$ 55 ($C_3H_3O^+$ and $C_4H_7^+$) serve as typical markers for tracing COA, as indicated in lines 306 and 307 of our manuscript.

In the context of our study, the inlet of the mobile laboratory was strategically positioned amidst the cooking plumes. Although we acknowledge that some degree of mixture between background air and vehicle emissions could occur, our primary assumption was that the majority of the emissions we collected were indeed COA.

References:

Chen, G., Canonaco, F., Tobler, A., Aas, W., Alastuey, A., Allan, J., Atabakhsh, S., Aurela, M., Baltensperger, U., Bougiatioti, A., De Brito, J. F., Ceburnis, D., Chazeau, B., Chebaicheb, H., Daellenbach, K. R., Ehn, M., El Haddad, I., Eleftheriadis, K., Favez, O., … Prévôt, A. S. H. (2022). European aerosol

*phenomenology − 8: Harmonised source apportionment of organic aerosol using 22 Year-long ACSM/AMS datasets. Environment International, 166, 107325. https://doi.org/10.1016/j.envint.2022.107325*

*Crippa, M., Canonaco, F., Lanz, V. A., Äijälä, M., Allan, J. D., Carbone, S., Capes, G., Ceburnis, D., Dall'Osto, M., Day, D. A., DeCarlo, P. F., Ehn, M., Eriksson, A., Freney, E., Hildebrandt Ruiz, L., Hillamo, R., Jimenez, J. L., Junninen, H., Kiendler-Scharr, A., … Prévôt, A. S. H. (2014). Organic aerosol components derived from 25 AMS data sets across Europe using a consistent ME-2 based source apportionment approach. Atmospheric Chemistry and Physics, 14(12), 6159–6176. https://doi.org/10.5194/acp-14-6159-2014*

5. The *m/z* above 100 should also be examined (such as *m/z* 109, 121, 131 and 135) which are present in fast food OA spectra for example.

Upon examining the m/z values you highlighted, we discovered a pattern in samples collected from the same sampling location. Specifically, both samples from the Bakery site exhibited a rapid increase around *m/z* 58 and *m/z* 86 (Figure S10). This can be attributed to the presence of nitrogen-containing fragments, namely $C_3H_8N^+$ (*m/z* 58) and $C_5H_{12}N^+$ (*m/z* 86), as discussed in our manuscript. An additional increase was observed around m/z 100, largely due to the significant contribution of $C_6H_{14}N^+$ at *m/z* 100.

In contrast, samples from other locations did not demonstrate such marked increases at specific *m/z* values. Instead, these samples exhibited a more gradual increase in the cumulative mass fraction, differing from the pattern observed in the Bakery samples. Consequently, we did not focus on the contributions of signals above 100 in these instances, as we did not identify a compelling necessity to do so given the minor role of these fragments in contributing to cooking emissions, based on our observations.

Tools like Positive Matrix Factorization (PMF) are great for pinpointing pollution sources. But in our study, we directly sampled emissions from restaurant cooking. This direct method made the usual PMF analysis less crucial because the source of emissions was clear. Using a mobile lab, we moved to various restaurants and took precise measurements. We kept a detailed log, noting any activities and changes during our sampling. Plus, we didn't just stop at collecting samples. We carried out thorough analyses on particle types and gases to give us a complete picture.

6. Regarding UFP, no losses in the chimney etc. have been taken into account or discussed, which can lead to misleading conclusions.

We are confident in our study's conclusions. While we recognize and account for potential losses of cooking emissions from the exhaust pipe, it's pivotal to emphasize that such losses likely affect emission concentrations primarily. Yet, we contend that these losses don't significantly alter the chemical profile of the emissions discussed in this manuscript. In other words, although emission quantities might change, their inherent chemical attributes, as indicated by our results, stay largely

unchanged. Hence, we firmly support our study's conclusions, rooted in meticulous observation and thorough analysis.

Minor comments:

7. Line 16. Aerodynamic or mobility diameter?

We've revised our manuscript's phrasing to better represent our methodology. Instead of the former 'particles <100 nm diameter', we now mention 'particles <100 nm mobility diameter' in line 16.

This adjustment is vital as it underscores that our particle size measurement is determined by the particle's behavior in an electric field, as measured by the Fast Mobility Particle Sizer (FMPS) utilized in our research. Such precision in our terminology guarantees a clearer understanding and accurate interpretation of our results.

8. Line 18. How much more elevated in comparison to urban background?

We've dedicated section '3.5. Particle size distributions and Ultrafine Particles (UFP) enhancements in restaurant plumes' to UFPs. In Figure 9, we show data adjusted for background effects, comparing these particles to urban background levels. As explained in the manuscript, these increases differ based on the restaurant site. Therefore, we didn't include an average number in the abstract. We aimed to discuss this in detail in the main text rather than simplifying it in the abstract.

9. Line 18. The majority of observed PM was organic aerosol (OA) by mass. Rephrase

We have revised the original sentence to enhance clarity and maintain professional tone in line 18 and 19, as follows:

" In our observations, Organic Aerosol (OA) predominantly constituted the particulate matter (PM) when assessed by mass.

10. Line 47. "have led to neighborhood-scale enhancements of ~0.5-1 µg m-3 of PM2.5". That's not such a high enhancement. This is not a strong motivation.

The National Ambient Air Quality Standard for $PM_{2.5}$ sets an annual limit of 12.0 µg/m³. Meanwhile, the World Health Organization recently lowered its recommendation to 5 µg/m³ from 10 µg/m³ in 2021. At first, this 5 µg/m³ might seem small. But, long-term health studies show that even small

amounts of PM$_{2.5}$ can be harmful. Consistent exposure to PM$_{2.5}$ increases heart and lung issues, and even the risk of lung cancer (Schwartz, 2000; Franklin et al., 2008). A U.S. study from 2000 to 2007 found that a 10 µg/m³ drop in PM$_{2.5}$ increased average lifespans by 0.35 years (Correia et al., 2013). This shows that even small changes in PM$_{2.5}$ can significantly impact health over time. So, a 0.5-1 µg/m³ increase in PM$_{2.5}$ is important. It highlights the need for our research and gives context to our results.

*References:*

*Schwartz, J. (2000). Harvesting and Long Term Exposure Effects in the Relation between Air Pollution and Mortality. American Journal of Epidemiology, 151(5), 440–448.*
*https://doi.org/10.1093/oxfordjournals.aje.a010228*
*Franklin, M., Koutrakis, P., & Schwartz, J. (2008). The Role of Particle Composition on the Association Between PM2.5 and Mortality. Epidemiology (Cambridge, Mass.), 19(5), 680–689.*
*Correia, A. W., Pope, C. A., Dockery, D. W., Wang, Y., Ezzati, M., & Dominici, F. (2013). The Effect of Air Pollution Control on Life Expectancy in the United States: An Analysis of 545 US counties for the period 2000 to 2007. Epidemiology (Cambridge, Mass.), 24(1), 23–31.*
*https://doi.org/10.1097/EDE.0b013e3182770237*

11. Lines 57-59. Referring that PM2.5 due to COA is 1.2 µg/m-3 in 2010, is not enough. What about 2020+? It could be even less. And what about PM1? OA due to traffic was 3 times higher than COA in 1982.

Our choice to contrast the two studies from 1982 and 2010 was driven by their shared sampling location in Pasadena, CA. This specific selection enabled us to examine changes in the contribution of cooking organic aerosols over time, using the same source apportionment model. As per your suggestion, we conducted a comprehensive search for more recent source apportionment studies conducted in the same location. However, the most recent research we found was the 2010 CalNex campaign in Pasadena, CA, which is already cited in our manuscript.

Our focus is on the robustness and relevance of the data rather than the recency, and we believe that the chosen studies provide a reliable and compelling analysis of the trends and shifts in cooking organic aerosols over time.

12. Line 59. "the fraction of urban PM2.5 attributed to cooking has increased". Reference?

The reference citation for the studies by Hayes et al., 2013, and Schauer et al., 1996, which provide context and details for the results discussed in the preceding sentence, have already been inserted in line 59 of the manuscript.

13. Line 63. Define that this is PM1 OA.

In accordance with your suggestion, we have revised the previous sentence to incorporate the clarification that the $PM_1$ referred to in the context is indeed PM1 Organic Aerosol (OA). The modified sentence is as follows:

Factor analysis utilizing the PMF (Positive Matrix Factorization) on AMS data routinely identifies a Cooking Organic Aerosol (COA) factor that represents between 6 - 25% of the total organic aerosol (OA) within $PM_1$ in urban settings.

14. Line 70. "While the UFPs from cooking can contribute to ~ 80% of the total particle Number". What about other sources? BBOA, HOA?

From the PMCAMx-UF data for summer in the eastern US, we found that biomass burning and dust makeup just 1% of ultrafine particle (UFP) emissions (Posner and Pandis, 2015). There's also a lack of research pointing to Biomass Burning Organic Aerosol (BBOA) in US cities. This makes sense, given that common sources like crop burning, wildfires, and wood heating are rare in urban areas. So, we believe biomass burning isn't a major source of UFPs in US cities.

The same study showed gasoline cars cause 40% of UFP emissions. But when most research mentions 'Hydrocarbon-like Organic Aerosol' (HOA), they use the Positive Matrix Factorization (PMF) model with $PM_{2.5}$ or NR-$PM_1$. So, UFPs from cars might not fit perfectly in the HOA group because UFPs are smaller than $PM_1$ or $PM_{2.5}$. This suggests that cars are a key source of UFPs. Meanwhile, BBOA seems to play a minor role, especially in the eastern US.

*References:*

*Posner, Laura N., and Spyros N. Pandis. "Sources of Ultrafine Particles in the Eastern United States." Atmospheric Environment 111 (June 1, 2015): 103–12. https://doi.org/10.1016/j.atmosenv.2015.03.033.*

15. Lines 101 and 145. CE=1. Why this choice was made?

The collection efficiency (CE) of elemental carbon (EC) values typically ranges between 0.5 and 1, contingent on the chemical composition and phase state of the aerosol. In the context of the capture vaporizer aerosol mass spectrometer (CV AMS), a collection efficiency of approximately 1 has been demonstrated for ambient aerosols (Hu et al., 2018), signifying complete (100%) efficiency in the capture vaporizer. Additionally, it's noteworthy that a majority of field studies employ a constant CE value for the quantification of ambient aerosols (Sun et al., 2010). We have further clarified this information in line 150-151.

*References:*

*Hu, Weiwei, Douglas A. Day, Pedro Campuzano-Jost, Benjamin A. Nault, Taehyun Park, Taehyoung Lee, Philip Croteau, et al. 2018. "Evaluation of the New Capture Vaporizer for Aerosol Mass Spectrometers (AMS): Elemental Composition and Source Apportionment of Organic Aerosols (OA)." ACS Earth and Space Chemistry 2 (4): 410–21. https://doi.org/10.1021/acsearthspacechem.8b00002.*

*Sun, Junying, Qi Zhang, Manjula R. Canagaratna, Yangmei Zhang, Nga L. Ng, Yele Sun, John T. Jayne, Xiaochun Zhang, Xiaoye Zhang, and Douglas R. Worsnop. 2010. "Highly Time- and Size-Resolved Characterization of Submicron Aerosol Particles in Beijing Using an Aerodyne Aerosol Mass Spectrometer." Atmospheric Environment 44 (1): 131–40. https://doi.org/10.1016/j.atmosenv.2009.03.020.*

16. Table 1. Define the type of diameter (AMS or SMPS). Explain if these values are average values from the total duration of sampling. I would suggest to add the average O:C ratios.

In Table 1, we have delineated that the Mode Dp is determined using the FMPS to specify the nature of the diameter. Furthermore, it should be noted that the average values presented were ascertained over the complete sampling duration for each respective location. In line with your valuable suggestion, we have incorporated the average O:C ratios for each restaurant, providing a more comprehensive view on the oxidation status and the contribution of SOA to the cooking emissions.

17. Table 1.  Data are not consistent to spectra. For example, for bakery 2, the f55 (0.003) does not coincide to f55 (0.02) in Figure S5.

We have corrected the numbers for f41, f43, and f55 in Table 1. Now, these data are consistent with Figure 4 and Figure S5.

18. Line 110. "As a result, the measured emission plumes went through varying degrees of dilution before reaching our sampling inlet". Please add how this dilution affected your results.

Our study addresses dilution implications in Section 4's "Conclusion and Atmospheric Relevance." In line 614, we highlight uncertainties from emission sources and dilution effects. We note that while the measured plumes experienced some dilution before sampling, the inlet's strategic position ensured that the OA's chemical attributes remained largely unaltered. However, dilution might have reduced UFPs count and OA concentration.

19. Line 194. "ΔOA and ΔBC were calculated by subtracting the background concentration from the measured OA or BC mass concentration." What about AMS OA spectra? Did you excluded the background spectrum? Please provide more info.

In line 201, we define the background concentration as the 5th percentile of the data from each sampling day. This approach, centered on percentiles of OA or BC concentrations, inherently precludes the generation of a background mass spectrum.

20. Line 198. "(ΔOA: 2.46 μg/m3)". It's not clear to what this refers to.

The unparenthesized value represents the mean OA concentration, whereas the value enclosed in parentheses (ΔOA: 2.46 μg/m$^3$) signifies the mean OA concentration after the subtraction of the background mass concentration.

21. Line 202. "enhancements in organic aerosol, black carbon, and PNC". Please provide a figure for these enhancements.

Regarding your request for a graphical representation of the enhancements, please refer to Figure 1 in our manuscript. This figure displays the data related to these enhancements, offering a clear visual summary. I trust Figure 1 will meet your needs. It depicts the complex interactions of these enhancements. We've ensured that the figure is thorough yet easy to understand.

22. Figure 1. I would prefer seeing the actual values and not ΔOA and ΔBC etc. I suggest including Fig. S3 in Figure 1.

We have included the background concentration details in Table S1. In accordance with your suggestion, we've also presented the raw OA and BC data, prior to the application of background level concentrations, along with the OA/BC ratio in Fig. S3.

23. Figure 2. The bar/Restaurant 2 box plot does not seem to be in accordance to data (Fig. S4).

We carefully looked at the data for bar/restaurant 2 in both Fig. S4 and Figure 2 and found them to be consistent. Figure 2 doesn't show some extreme values within the main range, which might make it seem different from Fig. S4 at first glance.

24. Figure 3. Please clarify in the manuscript, if the OA spectrum displays the average for the whole sampling period outside the Bar/Restaurant or only from spikes. The average OA spectrum for all experiments and not only for 3 datasets should be included in the SI.

We have emphasized the detail that Figure 3 represents the full sampling period at Bar/Restaurant 1 in Baltimore. In alignment with your recommendation, we've incorporated the average OA mass spectrum for all sampling locations in Figure S5, ensuring it encompasses both the latest sites and the preceding three.

25. Line 357. Fix the reference.

I have revised the sentence and updated the reference details in line 361 for greater clarity.

26. Line 361. Please define if you refer to bakery 1 or bakery 2? Or both? The same applies for fast food.

To provide a clearer understanding, I've adjusted the wording to 'both Bakery and Fast Food' in line 366.

27. Figure 5. In some cases R reveals a high correlation between the spectra obtained here and BBOA or even LV-OOA (Café, fast food 1 and 2). Why is this?

Our methodology focuses on similarities in mass spectra. Specifically, BBOA, representing biomass burning influenced Organic Aerosol, often displays $m/z$ 60 from the levoglucosan tracer and $m/z$ 73, typical in many biomass burnings. Referring to Crippa et al. (2013)'s BBOA mass spectrum, both $m/z$ 60 and $m/z$ 73 are evident. However, these markers aren't significantly higher than other signals like $m/z$ 28 and $m/z$ 44. The Café mass spectrum in Figure S5 shows strong contributions at $m/z$ 28 and $m/z$ 44, typical of oxidized OA, explaining the correlations seen between some cooking sites and LV-OOA. Due to real-world sampling, it's expected that some emissions will show varied oxidation levels.

[Figure]

28. Figure 5. The theta angle is also high between the mass spectra obtained here (values of 50+) implying no similarity between the different sites of this study. Have you compared the spectra from this study to more COA from literature? What about R^2?

Our samples from the Bakery site exhibit minimal similarity with other cooking sites and the COA factor as identified in Crippa et al. 2013. We have previously elaborated on the factors contributing to this limited similarity.

29. Line 389. "between mass spectra is larger than 0.8 and theta is less than 27°" This is not true. Many theta angles are higher than 30, 50 or even 70 (e.g. café, bakery1 and 2, Diner1, fast food 1). A theta angle of around 30° or even 20° also implies low similarity. Discuss.

In the discussed segment, I highlight that the majority of restaurants exhibit a correlation coefficient where R > 0.8 and θ < 27°. Out of 78 correlation values expressed as R, 35 exceed an R value of 0.8, accounting for approximately 45%. Regarding θ, 49 out of 78, or 63%, have values below 30°.

To eliminate any ambiguity, we've revised the previous statement as follows:

"For 45% of the restaurant sites studied, the correlation coefficient (R) for mass spectra exceeds 0.8. Concurrently, in the same representation, 63% of all restaurant sites exhibit a theta value of less than 30°. Collectively, these metrics underscore a notable similarity in mass spectra across a significant proportion of the sampling sites."

30. Line 398. "Correlations with BBOA and LVOOA are weaker". Not true for all cases. Rephrase.

In our study, referencing the findings of Crippa et al. (2013), we examined correlations with BBOA and LVOOA. Among our 13 restaurant sampling sites, 10 displayed an R value less than 0.5. Additionally, all the sampling sites exhibited a θ value exceeding 30°. This indicates that a significant portion of our restaurant samples have a low correlation with BBOA and LVOOA. The revised statement is as follows:

"In our study, the majority of restaurant sites exhibited weaker correlations with BBOA and LVOOA. This observation is attributed to the dominant peaks of BBOA at m/z 60 and 73 and of LVOOA at m/z 44 and 43."

31. Line 423. "CHN family peaks by mass, with significant contributions at *m/z* 41 and 43." There is also *m/z* 68, 79, and 83 for the specific spectrum. Please add discussion.

We have updated the sentence in line 429 incorporating further discussion:

In the mass spectrum presented for Bar/Restaurant 1 (Figure 3), the collective contribution of the CHN family peaks is quantified at 9.2% of the total signal mass. The nitrogen-containing fragment at $m/z$ 41, denoted as $CHN^+$, has a prominent 2.1% contribution. Subsequent significant peaks include $m/z$ 43 ($C_2H_5N^+$) at 0.77%, $m/z$ 79 ($C_5H_5N^+$) at 0.68%, and $m/z$ 68 ($C_4H_6N^+$) at 0.49%.

32. Line 429." 14 different nitrogen-containing". Are they present here? The results between this study and the reference show any similarity? Regarding amides etc., Booyens et al. (2019) for example states that the main source of nitriles, amides, and pyridine derivatives are most likely due to household combustion.

In our research, distinct CHN fragments identified from the AMS analysis included $CHN_2^+$, $C_2H_5N^+$, $CHNO^+$, $C_3H_5N^+$, and $C_2HNO^+$. Reyes-Villegas et al. (2018) reported several compounds in both gas and particle phases using FIGAERO-CIMS. Notably, $CHNO^+$ was the sole compound consistent between their findings and ours. It's imperative, however, to acknowledge the instrumental differences: the AMS, utilized in our study, measures fragments resulting from the hard ionization of the original compound, unlike the CIMS.

33. Line 439. "C3H8N+ (*m/z* 58) and C5H12N+ (*m/z* 86); together these two ions make up ~48% of the AMS-measured OA mass spectra". How did the average spectrum derived? Probably bakery should be treated differently than other restaurants (maybe a category of its own and not characterized as COA).

At each sampling location, such as Bakery 1, we computed the average mass spectrum over the sample collection duration. Subsequently, we determined the mass fraction contribution of this signal relative to the total signal intensity, which is standardized to 1.

In our manuscript, 'cooking emissions' encompass all emissions deriving from a restaurant's entire cooking cycle, from ingredient preparation to post-meal cleanup. Based on this definition, the Bakery site qualifies as a source of cooking emissions since it involves baking and directly offers the produced food items to patrons.

34. Line 440. "There is also a large contribution from C6H14N+ at $m/z$ 100." What about the contribution at $m/z$ 95, 97, 99? These summed up together are also contributing more than 10% of the OA. The bakery spectrum seems having no similarity at all, to typical COA spectra.

In the context of observed contributions at specific $m/z$ values for the Bakery spectra, we note the following distinctions: For Bakery 1, both $m/z$ 95 and 97, as well as $m/z$ 99, display a negligible CHN contribution, all registering at 0%. In contrast, $m/z$ 100 presents a more pronounced contribution of 1.64%, characterized by the presence of $C_6H_{14}N^+$. Turning our attention to Bakery 2, $m/z$ 95 yields a contribution of 0.50%, which is associated with $C_6H_9N^+$, while $m/z$ 97 exhibits a 0.70% contribution linked to $C_6H_{11}N^+$. Further, $m/z$ 99 offers a significant 1.44% contribution derived from $C_6H_{13}N^+$, and $m/z$ 100 stands out with a substantial 5.63% contribution from $C_6H_{14}N^+$.

When analyzing the bakery mass spectrum, it indeed exhibits distinct features compared to standard COA spectra. Yet, it's vital to note that these differences are primarily due to the nitrogen contributions at specific m/z values, rather than widespread deviations across the entire spectrum.

35. Line 444. "Barkery", should be bakery.

We have now modified the typographical error in the caption of Figure 6 in line 453.

36. Line 464. "As a result, we cannot conclude that the presence of ADA leads to high proportions of CHN ions" Bakery 1 and 2 are emissions from the same bakery. Could something irrelevant be emitted from the bakery chimney?

This statement highlights our uncertainty about the specific cooking processes at the bakery and the potential mixing with emissions from other sources. We aren't implying unrelated substances are emitted, but rather emphasizing the variability of the bakery's emissions.

37. Line 512. Define O/N

In line 521, I've defined the ratio 'O/N' to represent the proportion of Oxygen to Nitrogen.

38. Line 548. "This poor correlation may indicate that the emissions of OA and PNC". Have the authors taken into account losses of ultrafine particles in the chimney etc. until sampling?

We acknowledge potential emission losses, especially in ultrafine particles, as detailed in previous responses and the manuscript.

39. Figure 10. Are these data the measured ones or minus background? What corrections have been made (apart from normalization)?

In Figure 10, the displayed concentrations are before background subtraction. We first applied the CPC to FMPS correction, as outlined in section '2.2. Mobile laboratory and measurements', before normalizing the data.

40. Line 620. Fix the reference

The reference in the sentence has been revised in line 629 for clarity and to eliminate redundancy.

Comments on data and SI.

41. No data for gas-phase are shown for the rest of measurements.

In Figure S4, I have enriched the presentation by incorporating the available gas data alongside the organic and inorganic datasets.

42. Figure S4. I suggest including background and on-road data. Add on the top a relative time in h, e.g. define as time minus the background and on-road data, and as time 0, the start of sampling from restaurants. Move AMS OA on the right y-axis so that the rest AMS species' trends are visible. It would be easier to have captions for every place in the graph and not just on the figure description.

In Figure S4, I've incorporated the relative time above each illustration and designated a distinct Y-axis specifically for Organics, enhancing the visual clarity of organic concentrations. Additionally, I've annotated each figure with its respective sampling site name to bolster representation and comprehension.

43. Typical experiment is (k) (Bar/Restaurant 2). Why in fig. S4 OA are up to approx.. 2300 µg/m3 while in Fig. 1 1200 µg/m3? Even if you subtract the background (less than 5 µg/m3 of OA), 1000 µg/m3 difference is way high, making the data suspicious.

Thank you for pointing out the discrepancies between Figures S4 and Figure 1. The difference arises from the data processing methods used. Figure S4 displays raw AMS data recorded at 20-second intervals, while Figure 1 presents data averaged over one-minute intervals, with background corrections applied. This leads to the observed variance in concentration values.

44. Why g) and h) cases have elevated sulfate, ammonium mass conc. during certain periods?

The signals might represent the actual emission compositions at the sites. Despite this, figure g) and h) show that organics remain the predominant contributors at both sampling locations.

For example, while cooking can produce sulfate and ammonium, numerous indoor and outdoor sources might also influence their indoor concentrations. Elevated sulfate levels could stem from fuel combustion emitting sulfur dioxide ($SO_2$) or the consumption of sulfate-rich preserved foods. Similarly, ammonium levels during cooking might rise due to ammonia ($NH_3$) emissions from amino

acid breakdown, seafood preparation, or specific food additives. The use of ammonia-based cleaning products could also be a factor.

45. Figure S5. Other peaks that should be discussed are *m/z* 109, 121, 131 and 135.

As previously addressed in the accompanying figure concerning high mass fragments: samples from the Bakery site prominently exhibited elevated m/z values, particularly around *m/z* 58 and *m/z* 86. These peaks are indicative of nitrogen-containing fragments. Additionally, a marked peak was discernible at *m/z* 100, attributed to $C_6H_{14}N^+$. In contrast, samples from alternative locations didn't exhibit such pronounced increases, instead displaying a more uniform increment in the cumulative mass fraction. Consequently, for these non-Bakery samples, fragments with signals above 100 were not emphasized, considering their negligible influence on cooking emissions in our study.

46. The figures in SI do not appear in order. E.g. Figure S10 should be before Figure S8.

To enhance clarity and structure, we've created a specific section named '3. Offline sample preparation and analysis.' Within this section, Figure S13 (Figure S10 is now S13) is included. We've also refined the title to '2. LC-MS/MS Offline sample preparation and analysis' for a more concise representation and clearer understanding for our readers.

*References:*

*Booyens, W.; Van Zyl, P.G.; Beukes, J.P.; Ruiz-Jimenez, J.; Kopperi, M.; Riekkola, M.-L.; Vakkari, V.; Josipovic, M.; Kulmala, M.; Laakso, L. Characterising Particulate Organic Nitrogen at A Savannah-Grassland Region in South Africa. Atmosphere **2019**, 10, 492. https://doi.org/10.3390/atmos10090492.*

---

## Author Response (AR1)

**Point-by-point Response to Reviewer's Comments**

**Responses to Anonymous Referee #1**

• This review is about the manuscript Real-world observations of ultrafine particles and reduced nitrogen in commercial cooking organic aerosol emissions. The paper provides measurements of a wide range of aerosol properties using a variety of instruments. The main focus of the paper is the reduced nitrogen analyzed with a HR-AMS.

The paper is robust and can be published after addressing these comments.

We thank the reviewer for thoroughly reviewing our manuscript. We value the insights on the comprehensive measurements and the emphasis on the reduced nitrogen analyzed with a HR-AMS. We are committed to addressing the comments provided and believe that these revisions will further strengthen the paper. Below is our point-by-point response to each comment.

• My main concern is that the sample inlet (line 107) was a few meters from the exhaust vent. This suggests that the measurements will be mixed with ambient concentrations and also affected by wind speed and direction.

**Response:** We acknowledge the impact of dilution effects. We further highlighted the potential variability from ambient background air mixing in Line 119-125. Field campaigns inherently face unpredictability compared to controlled lab settings. Despite these challenges, they reflect real-world conditions. We delve deeper into these effects in Section 4, "Conclusion and Atmospheric Relevance."

**Revised manuscript** (Page 6, Line 119-125):**

Our procedure for identifying candidate restaurants has two important implications for our results. First, it means that the set of sampled restaurants represents a convenience sample and may therefore not be completely representative of the types of restaurants found in Pittsburgh or Baltimore. Second, since we did not coordinate with restaurant owners or operators during our sampling, we do not have detailed information about cooking fuel (though we assume that most restaurants used either gas or electricity), the specific cooking methods used, or the volume of food cooked during our sampling periods.

• Another concern is the fact that the authors measured in background, on-road and cooking environments at different times of the day and analysed the differences between sites, also calculating  $\Delta$ concentrations ( $\Delta$ OA and  $\Delta$ BC) based on the background concentrations. Please, explain why you can do this analysis.

**Response:** Using our mobile laboratory, we measured emissions from background, on-road, and cooking environments. While our initial plan didn't include assessing these diverse sites, driving the lab van to the cooking sample location inadvertently led us past the background site and several roads. This unintentional movement provided the opportunity to collect data from these extra sites.

Figure 1 graphically presents the differences in emission concentrations across these locations, highlighting the distinct characteristics of cooking emissions in contrast to the incidental background and road sites.

**Specific comments.**

• Line 44. Depending on the location, biomass burning is also a main OA source.

**Response:** We acknowledge biomass burning as a major source of Organic Aerosols (OA). Yet, our study specifically focuses on urban air pollution sources. We understand there might have been some ambiguity in our manuscript, leading to misconceptions.

To clarify, we've revised statements, notably in line 44, to emphasize our concentration on urban air quality in the United States.

**Revised manuscript** (Page 2, Line 44):**

Among a wide variety of contributing sources to air quality in the US, two notable urban sources are mobile sources (e.g., motor vehicles) and cooking.

• Line 59. The autors mention PM2.5 cooking has increase but the data used to make their point (line 56) states that their concentrations went from 2.4  $\mu$ g/m3 to 1.2  $\mu$ g/m3 56 between 1982 and 2010.

**Response:** Thank you for the kind comment. We have emphasized the increasing percentage of  $PM_{2.5}$  attributed to cooking. This may seem contradictory given the overall decrease in  $PM_{2.5}$  levels and reduced traffic emissions. However, it's important to clarify that while the absolute amount of  $PM_{2.5}$  from cooking has decreased, its relative contribution to the total  $PM_{2.5}$  has increased when compared to studies from 1982 and 2010. In essence, over time, cooking's proportionate share of  $PM_{2.5}$  has grown, even if its absolute contribution has lessened.

•Line 164. Are you sure that the AMS spectra with nitrogen-containing compounds are from the cooking exhaust?

**Response:** We adhered to methodologies endorsed by the AMS user group meeting to ensure the utmost precision in our data. To validate the presence of N-containing peaks in our mass spectrum, we examined the following steps:

First, one potential concern is that the formation of refractory components on the vaporizer surface can lead to conditioning that potentially affects the vaporizer interaction (Allan et al., 2003). However, after consultation with Aerodyne, it was confirmed that our signals were not due to surface ionization. Notably, peak shapes and peak widths of ions like Na+ and K+ were similar to Ar+, as seen in Figure S9. This similarity suggests consistent peak shapes without anomalies. If the signals from those metal elements show distinct lumpy or broad signals, it indicates that the ions are coming from

the surface of the vaporizer and follow a different path to get extracted into the MS, leading to different peak shapes.

Second, Figure S8 demonstrates the peak fitting of N-containing fragments, providing their genuine existence. Further assurance comes from the absence of signals when the chopper was closed, ruling out internal instrumental errors.

After thorough checks, we can assertively confirm our data's accuracy and eliminate the possibility of instrumental anomalies influencing the observed nitrogen contributions.

**References:**

Allan, J. D., Alfarra, M. R., Bower, K. N., Williams, P. I., Gallagher, M. W., Jimenez, J. L., ... & Worsnop, D. R. (2003). Quantitative sampling using an Aerodyne aerosol mass spectrometer 2. Measurements of fine particulate chemical composition in two UK cities. Journal of Geophysical Research: Atmospheres, 108(D3).

**Revised figures in supplementary** (Page 12-13):**

**1. Validation of Nitrogen-Containing Peak**

We addressed the potential issue of refractory components forming on the vaporizer surface, which could influence vaporizer interactions, as highlighted by Allan et al. (2003). Ions from surface ionization, generated at a different location than those from electron ionization, follow a distinct trajectory into the mass spectrometer, resulting in efficiency variances and shifted peaks in mass spectra (Drewnick et al., 2015). Additionally, distinctly irregular or expanded signals from metal ions (e.g., Na+ and K+) would imply their origin from the vaporizer surface, suggesting a divergent route for extraction into the MS.

To ascertain the presence of nitrogen-containing peaks in our mass spectrum and these fragments are not from surface ionization, we have confirmed that ions such as  $Na^+$  and  $K^+$  displayed peak configurations akin to Ar+, as illustrated in Figure S9. The consistency in peak shape and width shows that these ions are not from irregular signals. Subsequently, Figure S8 showcased the peak alignment of nitrogen-containing fragments, affirming their authentic presence. This was further corroborated by the nonexistence of signals during the chopper's inactivity, eliminating the possibility of internal instrumental discrepancies. Thus, our observation shows that our fragments seemingly do not emanate from the surface of the vaporizer.

**References:**

- Allan, J. D., Alfarra, M. R., Bower, K. N., Williams, P. I., Gallagher, M. W., Jimenez, J. L., ... & Worsnop, D. R. (2003). Quantitative sampling using an Aerodyne aerosol mass spectrometer 2. Measurements of fine particulate chemical composition in two UK cities. Journal of Geophysical Research: Atmospheres, 108(D3).
- Drewnick, F., Diesch, J.-M., Faber, P., & Borrmann, S. (2015). Aerosol mass spectrometry: Particle–vaporizer interactions and their consequences for the measurements. Atmospheric Measurement Techniques, 8(9), 3811–3830. https://doi.org/10.5194/amt-8-3811-2015

**Figure S8.** Example Peak Fitting at m/z 58 for Bakery 1 (left) and Bakery 2 (right): This demonstrates the minimal signal observed when the chopper was in the closed position.